# PLLay: Efficient Topological Layer based on Persistence Landscapes

**Kwangho Kim**
Carnegie Mellon University
Pittsburgh, USA
kwanghk@cmu.edu

**Jisu Kim**
Inria
Palaiseau, France
jisu.kim@inria.fr

**Manzil Zaheer**
Google Research
Mountain View, USA
manzilzaheer@google.com

**Joon Sik Kim**
Carnegie Mellon University
Pittsburgh, USA
joonsikk@cs.cmu.edu

**Frederic Chazal**
Inria
Palaiseau, France
frederic.chazal@inria.fr

**Larry Wasserman**
Carnegie Mellon University
Pittsburgh, USA
larry@stat.cmu.edu

## Abstract

We propose `PLLay`, a novel topological layer for general deep learning models based on persistence landscapes, in which we can efficiently exploit the underlying topological features of the input data structure. In this work, we show differentiability with respect to layer inputs, for a general persistent homology with arbitrary filtration. Thus, our proposed layer can be placed anywhere in the network and feed critical information on the topological features of input data into subsequent layers to improve the learnability of the networks toward a given task. A task-optimal structure of `PLLay` is learned during training via backpropagation, without requiring any input featurization or data preprocessing. We provide a novel adaptation for the DTM function-based filtration, and show that the proposed layer is robust against noise and outliers through a stability analysis. We demonstrate the effectiveness of our approach by classification experiments on various datasets.

## 1 Introduction

With its strong generalizability, deep learning has been pervasively applied in machine learning. To improve the learnability of deep learning models, various techniques have been proposed. Some of them have achieved an efficient data processing method through specialized layer structures; for instance, inserting a convolutional layer greatly improves visual object recognition and other tasks in computer vision [e.g., Krizhevsky et al., 2012, LeCun et al., 2016]. On the other hand, a large body of recent work focuses on optimal architecture of deep network [Simonyan and Zisserman, 2015, He et al., 2016, Szegedy et al., 2015, Albelwi and Mahmood, 2016].

In this paper, we explore an alternative way to enhance the learnability of deep learning models by developing a novel *topological layer* which feeds the significant topological features of the underlying data structure in an arbitrary network. The power of topology lies in its capacity which differentiates sets in topological spaces in a robust and meaningful geometric way [Carlsson, 2009, Ghrist, 2008]. It provides important insights into the global "shape" of the data structure via *persistent homology*

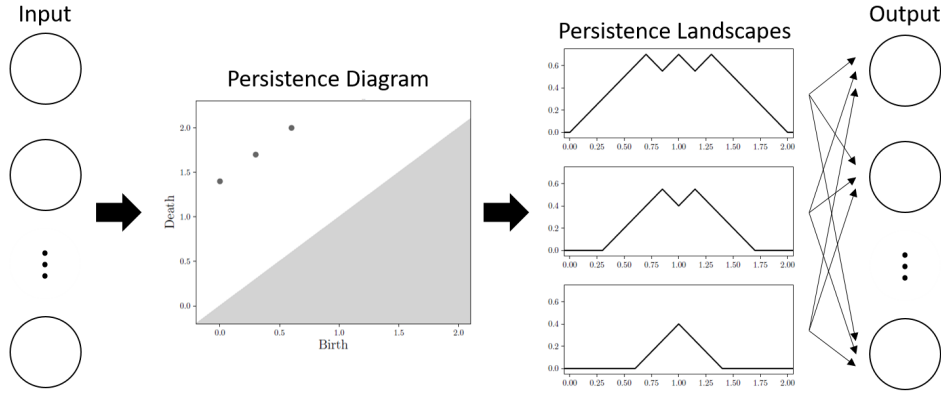

Figure 1: Illustration of `PLLay`, a novel topological layer based on weighted persistence landscapes. Information in the persistence diagram is first encoded into persistence landscapes as a form of vectorized function, and then a deep learning model determines which components of the landscape (e.g., particular hills or valleys) are important for a given task during training. `PLLay` can be placed anywhere in the network.

[Zomorodian and Carlsson, 2005]. The use of topological methods in data analysis has been limited by the difficulty of combining the main tool of the subject, persistent homology, with statistics and machine learning. Nonetheless, a series of recent studies have reported notable successes in utilizing topological methods in data analysis [e.g., Zhu, 2013, Dindin et al., 2020, Nanda and Sazdanović, 2014, Tralie and Perea, 2018, Seversky et al., 2016, Gamble and Heo, 2010, Pereira and de Mello, 2015, Umeda, 2017, Liu et al., 2016, Venkataraman et al., 2016, Emrani et al., 2014]

There are at least three benefits of utilizing the topological layer in deep learning; 1) we can efficiently extract robust global features of input data that otherwise would not be readily accessible via traditional feature maps, 2) an optimal structure of the layer for a given task can be easily embodied via backpropagation during training, and 3) with proper filtrations it can be applied to arbitrarily complicated data structure even without any data preprocessing.

**Related Work.** The idea of incorporating topological concepts into deep learning has been explored only recently, mostly via feature engineering perspective where we use some fixed, predefined features that contain topological information [e.g., Dindin et al., 2020, Umeda, 2017, Liu et al., 2016]. Guss and Salakhutdinov [2018], Rieck et al. [2019] proposed a complexity measure for neural network architectures based on topological data analysis. Carlsson and Gabrielsson [2020] applied topological approaches to deep convolutional networks to understand and improve the computations of the network. Hofer et al. [2017] first developed a technique to input persistence diagrams into neural networks by introducing their own topological layer. Carrière et al. [2020] proposed a network layer for persistence diagrams built on top of graphs. Poulenard et al. [2018], Gabrielsson et al. [2019], Hofer et al. [2019], Moor et al. [2020] also proposed various topology loss functions and layers applied to deep learning. Nevertheless, all the previous approaches suffer from at least one or more of the following limitations: 1) they rely on a particular parametrized map or filtration, 2) they lack stability results or the stability is limited to a particular type of input data representation, and 3) most importantly, the differentiability of persistent homology is not guaranteed with respect to the layer's input therefore we can not place the layer in the middle of deep networks in general.

**Contribution.** This paper presents a new topological layer, `PLLay` (Persistence Landscape-based topological Layer: see Figure 1 for an illustration), that does not suffer from the above limitations. Our topological layer does not rely on a particular filtration or a parametrized mapping but still shows favorable theoretical properties. The proposed layer is designed based on the weighted persistence landscapes to be less prone to extreme topological distortions. We provide a tight stability bound that does not depend on the input complexity, and show the stability with respect to input perturbations. We also provide a novel adaptation for the DTM function-based filtration, and analyze the stability property. Importantly, we guarantee the differentiability of our layer with respect to the layer's input.

**Reproducibility.** The code for `PLLay` is available at `https://github.com/jisuk1/pllay/`.

## 2    Background and definitions

*Topological data analysis* (TDA) is a recent and emerging field of data science that relies on topological tools to infer relevant features for possibly complex data [Carlsson, 2009]. In this section, we briefly review basic concepts and main tools in TDA which we will harness to develop our topological layer in this paper. We refer interested readers to Chazal and Michel [2017], Hatcher [2002], Edelsbrunner and Harer [2010], Chazal et al. [2009, 2016b] for details and formal definitions.

### 2.1    Simplicial complex, persistent homology, and diagrams

When inferring topological properties of $\mathbb{X}$, a subset of $\mathbb{R}^d$, from a finite collection of samples $X$, we rely on a *simplicial complex $K$*, a discrete structure built over the observed points to provide a topological approximation of the underlying space. Two common examples are the Čech complex and the Vietoris-Rips complex. The *Čech complex* is the simplicial complex where $k$-simplices correspond to the nonempty intersection of $k+1$ balls centered at vertices. The *Vietoris-Rips* (or simply *Rips*) *complex* is the simplicial complex where simplexes are built based on pairwise distances among its vertices. We refer to Appendix A for formal definitions.

A collection of simplicial complexes $\mathcal{F} = \{K_a \subset K : a \in \mathbb{R}\}$ satisfying $K_a \subset K_b$ whenever $a \leq b$ is called a *filtration* of $K$. A typical way of setting the filtration is through a monotonic function on the simplex. A function $f\colon K \to \mathbb{R}$ is monotonic if $f(\varsigma) \leq f(\tau)$ whenever $\varsigma$ is a face of $\tau$. If we let $K_a \coloneqq f^{-1}(-\infty, a]$, then the monotonicity implies that $K_a$ is a subcomplex of $K$ and $K_a \subset K_b$ whenever $a \leq b$. In this paper, we assume that the filtration is built upon a monotonic function.

*Persistent homology* is a multiscale approach to represent the topological features of the complex $K$, and can be represented in the persistence diagram. For a filtration $\mathcal{F}$ and for each nonnegative $k$, we keep track of when $k$-dimensional homological features (e.g., 0-dimension: connected component, 1-dimension: loop, 2-dimension: cavity,...) appear and disappear in the filtration. If a homological feature $\alpha_i$ appears at $b_i$ and disappears at $d_i$, then we say $\alpha_i$ is born at $b_i$ and dies at $d_i$. By considering these pairs $(b_i, d_i)$ as points in the plane, one obtains the *persistence diagram* defined as follows.

**Definition 2.1** *Let $\mathbb{R}_*^2 \coloneqq \{(b, d) \in (\mathbb{R} \cup \infty)^2 : d > b\}$. A persistence diagram $\mathcal{D}$ is a finite multiset of $\{p : p \in \mathbb{R}_*^2\}$. We let $\mathbb{D}$ denote the set of all such $\mathcal{D}$'s.*

We will use $\mathcal{D}_X, \mathcal{D}_{\mathbb{X}}$ as shorthand notations for the persistence diagram drawn from the simplicial complex constructed on original data source $X, \mathbb{X}$, respectively.

Lastly, we define the following metrics to measure the distance between two persistence diagrams.

**Definition 2.2 (Bottleneck and Wasserstein distance)**  *Given two persistence diagrams $\mathcal{D}$ and $\mathcal{D}'$, their bottleneck distance ($d_B$) and q-th Wasserstein distance ($W_q$) for $q \geq 1$ are defined by*

$$d_B(\mathcal{D}, \mathcal{D}') = \inf_{\gamma \in \Gamma} \sup_{p \in \bar{\mathcal{D}}} \|p - \gamma(p)\|_\infty, \qquad W_q(\mathcal{D}, \mathcal{D}') = \left[ \inf_{\gamma \in \Gamma} \sum_{p \in \bar{\mathcal{D}}} \|p - \gamma(p)\|_\infty^q \right]^{\frac{1}{q}}, \qquad (1)$$

*respectively, where $\|\cdot\|_\infty$ is the usual $L_\infty$-norm, $\bar{\mathcal{D}} = \mathcal{D} \cup Diag$ and $\bar{\mathcal{D}}' = \mathcal{D}' \cup Diag$ with Diag being the diagonal $\{(x, x) : x \in \mathbb{R}\} \subset \mathbb{R}^2$ with infinite multiplicity, and the set $\Gamma$ consists of all the bijections $\gamma\colon \bar{\mathcal{D}} \to \bar{\mathcal{D}}'$.*

Note that for all $q \in [1, \infty)$, $d_B(\mathcal{D}_X, \mathcal{D}_Y) \leq W_q(\mathcal{D}_X, \mathcal{D}_Y)$ for any given $\mathcal{D}_X, \mathcal{D}_Y$. As $q$ tends to infinity, the Wasserstein distance approaches the bottleneck distance. Also, see Appendix B for a further relationship between the bottleneck distance and Wasserstein distance.

### 2.2    Persistence landscapes

A persistence diagram is a multiset, which is difficult to be used as inputs for machine learning methods (due to the complicated space structure, cardinality issues, computationally inefficient metrics, etc.). Hence, it is useful to transform the persistent homology into a functional Hilbert space, where the analysis is easier and learning methods can be directly applied. One good example is the persistence landscape [Bubenik, 2015, 2018, Bubenik and Dłotko, 2017]. Let $\mathcal{D}$ denote a persistence

diagram that contains $N$ off-diagonal birth-death pairs. We first consider a set of piecewise-linear functions $\{\Lambda_p(t)\}_{p\in\mathcal{D}}$ for all birth-death pairs $p = (b,d) \in \mathcal{D}$ as

$$\Lambda_p(t) = \max\{0, \min\{t-b, d-t\}\}.$$

Then the *persistence landscape* $\lambda$ of the persistence diagram $\mathcal{D}$ is defined as a sequence of functions $\{\lambda_k\}_{k\in\mathbb{N}}$, where

$$\lambda_k(t) = \mathrm{kmax}_p\Lambda_p(t), \quad t \in \mathbb{R}, \ k \in \mathbb{N}, \tag{2}$$

Hence, the persistence landscape is a set of real-valued functions and is easily computable. Advantages for this kind of functional summaries are discussed in Chazal et al. [2014b], Berry et al. [2018].

## 2.3 Distance to measure (DTM) function

The Distance to measure (DTM) [Chazal et al., 2011, 2016a] is a robustified version of the distance function. More precisely, the DTM $d_{\mu,m_0}\colon \mathbb{R}^d \to \mathbb{R}$ for a probability distribution $\mu$ with parameter $m_0 \in (0,1)$ and $r \geq 1$ is defined by

$$d_{\mu,m_0}(x) = \left(\frac{1}{m_0}\int_0^{m_0}(\delta_{\mu,m}(x))^r dm\right)^{1/r},$$

where $\delta_{\mu,m}(x) = \inf\{t > 0 : \ \mu(\mathbb{B}(x,t)) > m\}$ when $\mathbb{B}(x,t)$ is an open ball centered at $x$ with radius $t$. If not specified, $r = 2$ is used as a default. In practice, we use a weighted empirical measure

$$P_n(x) = \frac{\sum_{i=1}^n \varpi_i \mathbb{1}(X_i = x)}{\sum_{i=1}^n \varpi_i},$$

with weights $\varpi_i$'s for $\mu$. In this case, we define the *empirical DTM* by

$$\hat{d}_{m_0}(x) = d_{P_n,m_0}(x) = \left(\frac{\sum_{X_i \in N_k(x)} \varpi_i' \left\|X_i - x\right\|^r}{m_0 \sum_{i=1}^n \varpi_i}\right)^{1/r}, \tag{3}$$

where $N_k(x)$ is the subset of $\{X_1,\ldots,X_n\}$ containing the $k$ nearest neighbors of $x$, $k$ is such that $\sum_{X_i \in N_{k-1}(x)} \varpi_i < m_0 \sum_{i=1}^n \varpi_i \leq \sum_{X_i \in N_k(x)} \varpi_i$, and $\varpi_i' = \sum_{X_j \in N_k(x)} \varpi_j - m_0 \sum_{j=1}^n \varpi_j$ if at least one of $X_i$'s is in $N_k(x)$ and $\varpi_i' = \varpi_i$ otherwise. Hence the empirical DTM behaves similarly to the $k$-nearest distance with $k = \lfloor m_0 n \rfloor$. For i.i.d cases, we typically set $\varpi_i = 1$ but the weights can be flexibly determined in data-driven way. The parameter $m_0$ determines how much topological/geometrical information should be extracted from the local or global structure. A brief guideline on DTM parameter selection can be found in Appendix F (see Chazal et al. [2011] for more details). Since the resulting persistence diagram is less prone to input perturbations and has nice stability properties, people often prefer using the DTM as their filtration function.

## 3 A novel topological layer based on weighted persistence landscapes

In this section, we present a detailed algorithm to implement PLLay for a general neural network. Let $X, \mathcal{D}_X, h_{\text{top}}$ denote our input, corresponding persistence diagram induced from $X$, the proposed topological layer, respectively. Broadly speaking, the construction of our proposed topological layer consists of two steps: 1) computing a persistence diagram from the input, and 2) constructing the topological layer from the persistence diagram.

### 3.1 Computation of diagram: $X \to \mathcal{D}_X$

To compute the persistence diagram from the input data, we first need to define the filtration which requires a simplicial complex $K$ and a function $f\colon K \to \mathbb{R}$. There are several options for $K$ and $f$. We are in general agnostic about which filtration to use since it is in fact problem-dependent; in practice, we suggest using ensemble-like methods that can adapt to various underlying topological structures. One popular choice is the Vietoris-Rips filtration. When there is a one-to-one correspondence between $X_i$ and each fixed grid point $Y_i$, one obvious choice for $f$ could be just interpreting $X$ as a function values, so $f(Y_i) = X_i$. We refer to Chazal and Michel [2017] for more examples.

As described in Section 2.3, one appealing choice for $f$ is the DTM function. Due to its favorable properties, the DTM function has been widely used in TDA [Anai et al., 2019, Xu et al., 2019], and

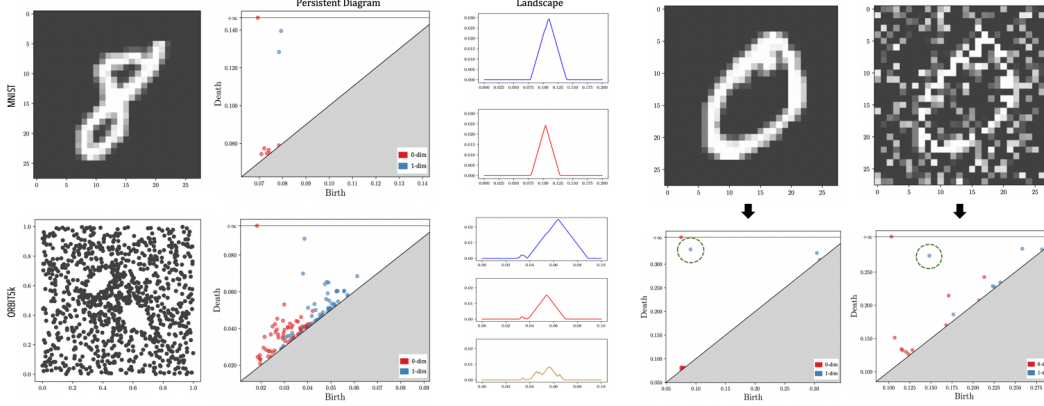

Figure 2: The topological features encoded in the persistence diagram & persistence landscapes for MNIST and ORBIT5k sample. In the MNIST example, two loops (1-dimensional feature) in '8' are clearly identified and encoded into the 1st and 2nd order landscapes. The ORBIT5k sample shows more involved patterns.

Figure 3: The significant point (inside green-dashed circle) in the persistence diagram remains almost unchanged even after corrupting pixels and adding noise to the image.

has a good potential for deep learning application. Nonetheless, to the best of our knowledge, the DTM function has not yet been adopted in previous studies. In what follows, we detail two common scenarios for the DTM adaptation: when we consider the input $X$ as 1) data points or 2) weights.

- If the input data $X$ is considered as the empirical data points, then the empirical DTM in (3) with weights $\varpi_i$'s becomes

$$\hat{d}_{m_0}(x) = \left( \frac{\sum_{X_i \in N_k(x)} \varpi_i' \, \|X_i - x\|^r}{m_0 \sum_{i=1}^n \varpi_i} \right)^{1/r},$$
(4)

where $k$ and $\varpi_i'$ are determined as in (3).

- If the input data $X$ is considered as the weights corresponding to fixed points $\{Y_1, \dots, Y_n\}$, then the empirical DTM in (3) with data points $Y_i$'s and weights $X_i$'s becomes

$$\hat{d}_{m_0}(x) = \left( \frac{\sum_{X_i \in N_k(x)} X_i' \, \|Y_i - x\|^r}{m_0 \sum_{i=1}^n X_i} \right)^{1/r},$$
(5)

where $k$ and $\varpi_i'$ are determined as in (3).

Figure 2 provides some real data examples (which will be used in Section 5) of the persistence diagrams and the corresponding persistence landscapes based on the DTM functions. As shown in Figure 3, the topological features are expected to be robust to external noise or corruption.

### 3.2 Construction of topological layer: $\mathcal{D}_X \to h_{\mathbf{top}}$

Our topological layer is defined based on a parametrized mapping which takes the persistence diagram $\mathcal{D}$ to be projected onto $\mathbb{R}$, by harnessing persistence landscapes. Our construction is less afflicted by the artificial bending due to a particular transformation procedure as in Hofer et al. [2017], yet still guarantees the crucial information in the persistence diagram to be well preserved as will be seen in Section 4. Insignificant points with low persistence are likely to be ignored systematically without introducing additional nuisance parameters [Bubenik and Dłotko, 2017].

Let $\mathbb{R}^{+0}$ denote $[0, \infty)$. Given a persistence diagram $\mathcal{D} \in \mathbb{D}$, we compute the persistence landscape of order $k$ in (2), $\lambda_k(t)$, for $k = 1, ..., K_{max}$. Then, we compute the weighted average $\overline{\lambda}_{\boldsymbol{\omega}}(t) := \sum_{k=1}^{K_{max}} \omega_k \lambda_k(t)$ with a weight parameter $\boldsymbol{\omega} = \{\omega_k\}_k$, $\omega_k > 0$, $\sum_k \omega_k = 1$. Next, we set a domain $[T_{\min}, T_{\max}]$ and a resolution $\nu := T/(m-1)$, and sample $m$ equal-interval points from $[T_{\min}, T_{\max}]$ to obtain $\overline{\boldsymbol{\Lambda}}_{\boldsymbol{\omega}} = \left( \overline{\lambda}_{\boldsymbol{\omega}}(T_{\min}), \overline{\lambda}_{\boldsymbol{\omega}}(T_{\min} + \nu), ..., \overline{\lambda}_{\boldsymbol{\omega}}(T_{\max}) \right)^{\top} \in \left( \mathbb{R}^{+0} \right)^m$. Consequently, we have defined a mapping $\overline{\boldsymbol{\Lambda}}_{\boldsymbol{\omega}} : \mathbb{D} \to \left( \mathbb{R}^{+0} \right)^m$ which is a (vectorized) finite-sample approximation of the weighted persistence landscapes at the resolution $\nu$, at fixed, predetermined locations. Finally, we consider a parametrized differentiable map $g_{\boldsymbol{\theta}} : \left( \mathbb{R}^{+0} \right)^m \to \mathbb{R}$ which takes the input $\overline{\boldsymbol{\Lambda}}_{\boldsymbol{\omega}}$ and is differentiable with respect to $\boldsymbol{\theta}$ as well. Now, the projection of $\mathcal{D}$ with respect to the mapping

---

**Algorithm 1** Implementation of single structure element for PLLay

**Input:** persistence diagram $\mathcal{D} \in \mathbb{D}$

1. compute $\lambda_k(t)$ (2) on $t \in [0, T]$ for every $k = 1, ..., K_{max}$

2. compute the weighted average $\overline{\lambda}_{\boldsymbol{\omega}}(t) := \sum_{k=1}^{K_{max}} \omega_k \lambda_k(t)$, $\omega_k > 0$, $\sum_k \omega_k = 1$

3. set $\nu := \frac{T}{m-1}$, and compute $\overline{\boldsymbol{\Lambda}}_{\boldsymbol{\omega}} = (\overline{\lambda}_{\boldsymbol{\omega}}(T_{\min}), \overline{\lambda}_{\boldsymbol{\omega}}(T_{\min} + \nu), ..., \overline{\lambda}_{\boldsymbol{\omega}}(T_{\max}))^\top \in \mathbb{R}^m$

4. for a parametrized differentiable map $g_{\boldsymbol{\theta}} \colon \mathbb{R}^m \to \mathbb{R}$, define $S_{\boldsymbol{\theta}, \boldsymbol{\omega}} = g_{\boldsymbol{\theta}} \circ \overline{\boldsymbol{\Lambda}}_{\boldsymbol{\omega}}$

**Output:** $S_{\boldsymbol{\theta}, \boldsymbol{\omega}} \colon \mathbb{D} \to \mathbb{R}$

---

$S_{\boldsymbol{\theta}, \boldsymbol{\omega}} := g_{\boldsymbol{\theta}} \circ \overline{\boldsymbol{\Lambda}}_{\boldsymbol{\omega}}$ defines a single *structure element* for our topological input layer. We summarize the procedure in Algorithm 1.

The projection $S_{\boldsymbol{\theta}, \boldsymbol{\omega}}$ is continuous at every $t \in [T_{\min}, T_{\max}]$. Also, note that it is differentiable with respect to $\boldsymbol{\omega}$ and $\boldsymbol{\theta}$, regardless of the resolution level $\nu$. In what follows, we provide some guidelines that might be useful to implement Algorithm 1.

$\boldsymbol{\omega}$: The weight parameter $\boldsymbol{\omega}$ can be initialized uniformly, i.e. $\omega_k = 1/K_{max}$ for all $k$, and will be re-determined during training through the softmax layer in a way that a certain landscape conveying significant information has more weight. In general, lower-order landscapes tend to be more significant than higher-order landscapes, but the optimal weights may vary from task to task.

$\boldsymbol{\theta}, \mathbf{g}_{\boldsymbol{\theta}}$: Likewise, some birth-death pairs, encoded in the landscape function, may contain more crucial information about the topological features of the input data structure than others. Roughly speaking, this is equivalent to say certain mountains (or their ridge or valley) in the landscape are especially important. Hence, the parametrized map $g_{\boldsymbol{\theta}}$ should be able to reflect this by its design. In general, it can be done by affine transformation with scale and translation parameter, followed by an extra nonlinearity and normalization if necessary. We list two possible choices as below.

- Affine transformation: with scale and translation parameter $\boldsymbol{\sigma}_i, \boldsymbol{\mu}_i \in \mathbb{R}^m$, $g_{\boldsymbol{\theta}_i}(\overline{\boldsymbol{\Lambda}}_{\boldsymbol{\omega}}) = \boldsymbol{\sigma}_i^\top(\overline{\boldsymbol{\Lambda}}_{\boldsymbol{\omega}} - \boldsymbol{\mu}_i)$ and $\boldsymbol{\theta}_i = (\boldsymbol{\sigma}_i, \boldsymbol{\mu}_i)$.

- Logarithmic transformation: with same $\boldsymbol{\theta}_i = (\sigma_i, \boldsymbol{\mu}_i)$, $g_{\boldsymbol{\theta}_i}(\overline{\boldsymbol{\Lambda}}_{\boldsymbol{\omega}}) = \exp\left(-\sigma_i \|\overline{\boldsymbol{\Lambda}}_{\boldsymbol{\omega}} - \boldsymbol{\mu}_i\|_2\right)$.

Note that other constructions of $g_{\boldsymbol{\theta}}, \boldsymbol{\theta}, \boldsymbol{\omega}$ are also possible as long as they satisfy the sufficient conditions described above. Finally, since each structure element corresponds to a single node in a layer, we concatenate many of them, each with different parameters, to form our topological layer.

**Definition 3.1 (Persistence landscape-based topological layer (PLLay))** *For $n_h \in \mathbb{N}$, let $\boldsymbol{\eta}_i = (\boldsymbol{\theta}_i, \boldsymbol{\omega}_i)$ denote the set of parameters for the $i$-th structure element and let $\boldsymbol{\eta} = (\boldsymbol{\eta}_i)_{i=1}^{n_h}$. Given $\mathcal{D}$ and resolution $\nu$, we define PLLay as a parametrized mapping with $\boldsymbol{\eta}$ of $\mathbb{D} \to \mathbb{R}^{n_h}$ such that*

$$h_{top} \colon \mathcal{D} \to \left(S_{\boldsymbol{\eta}_i}(\mathcal{D}; \nu)\right)_{i=1}^{n_h}. \tag{6}$$

Note that this is nothing but a concatenation of $n_h$ topological structure elements (nodes) with different parameter sets (thus $n_h$ is our layer dimension).

**Remark 1** *Our PLLay considers only $K_{\max}$ top landscape functions. For a given persistence diagram, the points near the diagonal are not likely to appear at $K_{\max}$ top landscape functions, and hence not considered in PLLay. And hence PLLay automatically filters out the noisy features.*

### 3.3 Differentiability

This subsection is devoted to the analysis of the differential behavior of PLLay with respect to its input (or output from the previous layer), by computing the derivatives $\frac{\partial h_{top}}{\partial X}$. Since $\frac{\partial h_{top}}{\partial X} = \frac{\partial h_{top}}{\partial \mathcal{D}_X} \circ \frac{\partial \mathcal{D}_X}{\partial X}$, this can be done by combining two derivatives $\frac{\partial \mathcal{D}_X}{\partial X}$ and $\frac{\partial h_{top}}{\partial \mathcal{D}_X}$. We have extended Poulenard et al. [2018] so that we can compute the above derivatives for general persistent homology under arbitrary filtration in our setting. We present the result in Theorem 3.1.

**Theorem 3.1** *Let $f$ be the filtration function. Let $\xi$ be a map from each birth-death point $(b_i, d_i) \in \mathcal{D}_X$ to a pair of simplices $(\beta_i, \delta_i)$. Suppose that $\xi$ is locally constant at $X$, and $f(\beta_i)$ and $f(\delta_i)$ are differentiable with respect to $X_j$'s. Then, $h_{top}$ is differentiable with respect to $X$ and*

$$\frac{\partial h_{top}}{\partial X_j} = \sum_i \frac{\partial f(\beta_i)}{\partial X_j} \sum_{l=1}^m \frac{\partial g_{\boldsymbol{\theta}}}{\partial x_l} \sum_{k=1}^{K_{\max}} \omega_k \frac{\partial \lambda_k(lv)}{\partial b_i} + \sum_i \frac{\partial f(\delta_i)}{\partial X_j} \sum_{l=1}^m \frac{\partial g_{\boldsymbol{\theta}}}{\partial x_l} \sum_{k=1}^{K_{\max}} \omega_k \frac{\partial \lambda_k(lv)}{\partial d_i}.$$

The proof is in Appendix E.1. Note that $\frac{\partial \lambda_k}{\partial b_i}, \frac{\partial \lambda_k}{\partial d_i}$ are piecewise constant and are easily computed in explicit forms. Also $\frac{\partial g_{\boldsymbol{\theta}}}{\partial x_l}$ can be easily realized by an automatic differentiation framework such as `tensorflow` or `pytorch`. Our `PLLay` in Definition 3.1 is thus trainable via backpropagation at an arbitrary location in the network. In Appendix D, we also provide a derivative for the DTM filtration.

## 4 Stability Analysis

A key property of `PLLay` is stability; its discriminating power should remain stable against non-systematic noise or perturbation of input data. In this section, we shall provide our theoretical results on the stability properties of the proposed layer. We first address the stability for each structure element with respect to changes in persistence diagrams in Theorem 4.1.

**Theorem 4.1** *Let $g_{\boldsymbol{\theta}}$ be $\| \cdot \|_\infty$-Lipschitz, i.e. there exists $L_g > 0$ with $|g_{\boldsymbol{\theta}}(x) - g_{\boldsymbol{\theta}}(y)| \leq L_g \|x - y\|_\infty$ for all $x, y \in \mathbb{R}^m$. Then for two persistence diagrams $\mathcal{D}, \mathcal{D}'$,*
$$|S_{\boldsymbol{\theta},\boldsymbol{\omega}}(\mathcal{D}; \nu) - S_{\boldsymbol{\theta},\boldsymbol{\omega}}(\mathcal{D}'; \nu)| \leq L_g d_B(\mathcal{D}, \mathcal{D}').$$

Proof of Theorem 4.1 is given in Appendix E.2. Theorem 4.1 shows that $S_{\boldsymbol{\theta},\boldsymbol{\omega}}$ is stable with respect to perturbations in the persistence diagram measured by the bottleneck distance (1). It should be noted that only the Lipschitz continuity of $g_{\boldsymbol{\theta}}$ is required to establish the result.

Next, Corollary 4.1 shows that under certain conditions our approach improves the previous stability result of Hofer et al. [2017].

**Corollary 4.1** *For $t > 0$, let $n_t \in \mathbb{N}$ be satisfying that, for any two diagrams $\mathcal{D}_t, \mathcal{D}'_t$ with $d_B(\mathcal{D}, \mathcal{D}_t) \leq t$ and $d_B(\mathcal{D}', \mathcal{D}'_t) \leq t$, either $\mathcal{D}_t \backslash \mathcal{D}'_t$ or $\mathcal{D}'_t \backslash \mathcal{D}_t$ has at least $n_t$ points. Then, the ratio of our stability bound in Theorem 4.1 to that in Hofer et al. [2017] is upper bounded by*
$$C_{g_\theta}/(1 + (2t/d_B(\mathcal{D}, \mathcal{D}')) \times (n_t - 1)),$$
*where $C_{g_\theta}$ is a constant to be specified in the proof.*

See Appendix E.3 for the proof. Corollary 4.1 implies that for complex data structures where each $\mathcal{D}$ contains many birth-death pairs (for fixed $t$, in general $n_t$ grows with the increase in the number of points in $\mathcal{D}$), our stability bound is tighter than that of Hofer et al. [2017] at polynomial rates.

In particular, when we use the DTM function-based filtration proposed in (4) and (5), Theorem 4.1 can be turned into the following stability result with respect to our input $X$.

**Theorem 4.2** *Suppose $r = 2$ is used for the DTM function. Let a differentiable function $g_{\boldsymbol{\theta}}$ and resolution $\nu$ be given, and let $P$ be a distribution. For the case when $X_j$'s are data points, i.e. when (4) is used as the DTM function of $X$, let $P_n$ be the empirical distribution defined by $P_n = \frac{\sum_{i=1}^n \varpi_i \delta_{X_i}}{\sum_{i=1}^n \varpi_i}$. For the case when $X_j$'s are weights, i.e. when (5) is used as the DTM function of $X$, let $P_n$ be the empirical distribution defined by $P_n = \frac{\sum_{i=1}^n X_i \delta_{Y_i}}{\sum_{i=1}^n X_i}$. Let $\mathcal{D}_P$ be the persistence diagram of the DTM filtration of $P$, and $\mathcal{D}_X$ be the persistence diagram of the DTM filtration of $X$. Then,*
$$|S_{\boldsymbol{\theta},\boldsymbol{\omega}}(\mathcal{D}_X; \nu) - S_{\boldsymbol{\theta},\boldsymbol{\omega}}(\mathcal{D}_P; \nu)| \leq L_g m_0^{-1/2} W_2(P_n, P).$$

The proof is given in Appendix E.4. Theorem 4.2 implies that if the empirical distribution $P_n$ induced from the given input $X$ well approximates the true distribution $P$ with respect to the Wasserstein distance, i.e. having small $W_2(P_n, P)$, then `PLLay` constructed on observed data is close to the one as if we were to know the true distribution $P$.

Theorem 4.1 and 4.2 suggest that the topological information embedded in the proposed layer is robust against small noise, data corruption, or outliers. We have also discussed the stability result for the Vietoris-Rips and the Čech complex in Appendix C.

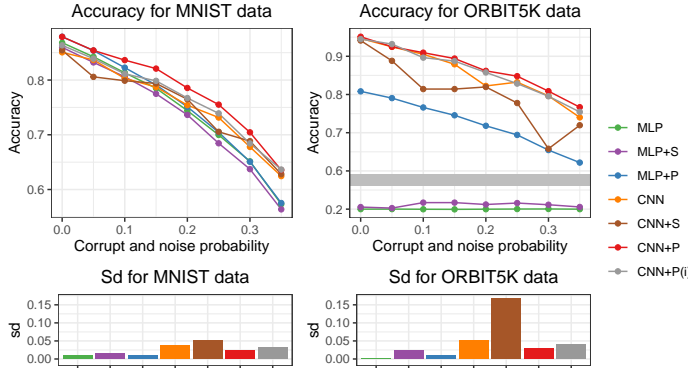

| Model | Accuracy |
|---|---|
| PointNet | 0.708 |
| | ($\pm 0.285$) |
| PersLay | 0.877 |
| | ($\pm 0.010$) |
| CNN | 0.915 |
| | ($\pm 0.088$) |
| CNN+ | 0.943 |
| SLay | ($\pm 0.014$) |
| **CNN+** | **0.950** |
| **PLLay** | **($\pm 0.016$)** |

Figure 4: Test accuracy in `MNIST` and `ORBIT5K` experiments. `PLLay` consistently improves the accuracy and the robustness against noise and corruption. In particular, in many cases it effectively reduces the variance of the classification accuracy on `ORBIT5K`.

Table 1: Comparison of different methods for `ORBIT5K` including the current state-of-the-art PersLay. The proposed method achieves the new state-of-the-art accuracy.

## 5   Experiments

To demonstrate the effectiveness of the proposed approach, we study classification problems on two different datasets: `MNIST` handwritten digits and `ORBIT5K`. To fairly showcase the benefits of using our proposed method, we keep our network architecture as simple as possible so that we can focus on the contribution from `PLLay`. In the experiments, we aim to explore the benefits of our layer through the following questions: 1) does it make the network more robust and reliable against noise, etc.? and 2) does it improve the overall generalization capability compared to vanilla models? In order to address both of these questions, we first consider the *corruption process*, a certain amount of random omission of pixel values or points from each raw example (so we will have less information), and the *noise process*, a certain amount of random addition of uniformly-distributed noise signals or points to each raw example. An example is given in Figure 3. Then we fit a standard multilayer perceptron (MLP) and a convolutional neural network (CNN) with and without the augmentation of `PLLay` across various noise and corruption rates given to the raw data, and compare the results. The guideline for choosing the TDA parameters in this experiment is described in Appendix F. We intentionally use a small number of training data ($\sim 1000$) so that the convergence rates could be included in the evaluation criteria. Each simulation is repeated 20 times. We refer to Appendix G for details about each simulation setup and our model architectures.

**MNIST handwritten digits**

We classify handwritten digit images from `MNIST` dataset. Each digit has distinctive topological information which can be encoded into the Persistence Landscape as in Figure 2.

**Topological layer.** We add two parallel `PLLay`s in Definition 6 at the beginning of MLP and CNN models, based on the empirical DTM function in (5), where we define fixed $28 \times 28$ points on grid and use a set of grayscale values $X$ as a weight vector for the fixed points. We used $m_0 = 0.05$ and $m_0 = 0.2$ for each layer, respectively (referred to MLP+P, CNN+P(i), respectively). Particularly for the CNN model, it is likely that the output of the convolutional layers might carry significant information about (smoothed) geometry of the input data shape. So we additionally place another `PLLay` after each convolutional layer, directly taking the layer output as 2D-function values and using the sublevel filtration (CNN+P).

**Baselines.** As our baseline methods, we employ 2-layer vanilla MLP, 2-layer CNN, and the topological signature method by Hofer et al. [2017] based on the empirical DTM function proposed in (5) (which we will refer to as SLay). The SLay is augmented at the beginning of MLP and CNN, referred to as MLP+S and CNN+S. See Appendix G.1 for more details.

**Result.** In Figure 4, we observe that `PLLay` augmentation consistently improves the accuracy of all the baselines. Interestingly, as we increase the corruption and noise rates, the improvement on

CNN increases up to the moderate level of corruption and noise ($\sim 15\%$), then starts to decrease. We conjecture that this is because although DTM filtration is able to robustly capture homological signals as illustrated in Figure 2, if the corruption and noise levels become too much, then the topological structure starts to dissolve in the DTM filtration.

### Orbit Recognition

We classify point clouds generated by 5 different dynamical systems from `ORBIT5K` dataset [Adams et al., 2017, Carrière et al., 2020]. The detailed data generating process is described in Appendix G.2.

**Topological layer.** The setup remains the same as in the previous `MNIST` case, except that 1) `PLLay` at the beginning of each network uses the empirical DTM function in (4), and 2) we set $m_0 = 0.02$.

**Baselines & Simulation.** All the baseline methods remain the same. For noiseless case, we added PointNet [Charles et al., 2017], a state-of-the-art in point cloud classification, and PersLay [Carrière et al., 2020], a state-of-the-art in TDA-utilized classification.

**Result.** In Figure 4, we observe that `PLLay` improves upon MLP and MLP+S by a huge margin ($42\% \sim 60\%$). In particular, without augmenting `PLLay`, MLP and MLP+S remain at almost a random classifier, which implies that the topological information is indeed crucial for the `ORBIT5K` classification task, and it would otherwise be very challenging to extract meaningful features. `PLLay` improves upon CNN or CNN+S consistently as well. Moreover, it appears that CNN suffers from high variance due to the high complexity of `ORBIT5K` dataset. On the other hand, `PLLay` can effectively mitigate this problem and make the model more stable by utilizing robust topological information from DTM function. Impressively, for the noiseless case, `PLLay` has achieved better performance than all the others including the current state-of-the-art PointNet and PersLay by a large margin.

## 6 Discussion

In this study, we have presented `PLLay`, a novel topological layer based on the weighted persistence landscape where we can exploit the topological features effectively. We provide the differentiability guarantee of the proposed layer with respect to the layer's input under arbitrary filtration. Hence, our study offers the first general topological layer which can be placed anywhere in the deep learning network. We also present new stability results that verify the robustness and efficiency of our approach. It is worth noting that our method and analytical results in this paper can be extended to silhouettes [Chazal et al., 2015, 2014b]. In the experiments, we have achieved the new state-of-the-art accuracy for `ORBIT5K` dataset based on the proposed method. We expect our work to bridge the gap between modern TDA tools and deep learning research.

The computational complexity depends on how `PLLay` is used. Computing the DTM is $O(n + m \log n)$ when $m_0 \propto 1/n$ and k-d tree is used, where $n$ is the input size and $m$ is the grid size. Computing the persistence diagram is $O(m^{2+\epsilon})$ for any small $\epsilon > 0$ when the simplicial complex $K$ in Section 3.1 grows linearly with respect to the grid size such as cubical complex or alpha complex (Chen and Kerber [2013] and Theorem 4.4, 5.6 of Boissonnat et al. [2018]). Computing the persistence landscape grows linearly with respect to the number of homological features in the persistence diagram, which is the topological complexity of the input and does not necessarily depend on $n$ or $m$. For our experiments, we consider fixed grids of size $28 \times 28$ and $40 \times 40$ as in Appendix G, so the computation is not heavy. Also, if we put `PLLay` only at the beginning of the deep learning model, then `PLLay` can be pre-computed and needs not to be calculated at every epoch in the training.

There are several remarks regarding our experiments. First, we emphasize that SLay in Section 5 is rather an intermediate tool designed for our simulation and not completely identical to the topological signature method by Hofer et al. [2017]. For example, SLay combines the method by Hofer et al. [2017] and the DTM function in (4) and (5) that have not appeared in the previous study. So we cannot exclude the possibility that the comparable performance of SLay for certain simulations is due to the contribution by the DTM function filtration. Moreover, for CNN, placing extra `PLLay` after each convolutional layer appears to bring marginal improvement in accuracy in our experiments. Exploring the optimal architecture with our `PLLay`, e.g., finding the most accurate and efficient `PLLay` network for a given classification task, would be an interesting future work.

The source code of `PLLay` is publicly available at `https://github.com/jisuk1/pllay/`.

## Broader Impact

This paper proposes a novel method of adapting tools in applied mathematics to enhance the learnability of deep learning models. Even though our methodology is generally applicable to any complex modern data, it is not tuned to a specific application that might improperly incur direct societal/ethical consequences. So the broader impact discussion is not needed for our work.

## Acknowledgments and Disclosure of Funding

During the last 36 months prior to the submission, Jisu Kim received Samsung Scholarship, and Joon Sik Kim received Kwanjeong Fellowship. Freédéric Chazal was supported by the ANR AI chair TopAI.

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
