[Supplementary Material]

# APPENDIX

## A    Simplicial complex, Persistent homology, and Distance between sets on metric spaces

Throughout, we will let $\mathbb{X}$ denotes a subset of $\mathbb{R}^d$, and $X$ denotes a finite collection of points from an arbitrary space $\mathbb{X}$.

A simplicial complex can be seen as a high dimensional generalization of a graph. Given a set $V$, an *(abstract) simplicial complex* is a set $K$ of finite subsets of $V$ such that $\alpha \in K$ and $\beta \subset \alpha$ implies $\beta \in K$. Each set $\alpha \in K$ is called its *simplex*. The *dimension* of a simplex $\alpha$ is $\dim \alpha = \mathrm{card}\,\alpha - 1$, and the dimension of the simplicial complex is the maximum dimension of any of its simplices. Note that a simplicial complex of dimension 1 is a graph.

When approximating the topology of the underlying space by observed samples, a common choice is the *Čech complex,* defined next. Below, for any $x \in \mathbb{X}$ and $r > 0$, we let $\mathbb{B}_{\mathbb{X}}(x, r)$ denote the open ball centered at $x$ and radius $r > 0$ intersected with $\mathbb{X}$.

**Definition A.1 (Čech complex)** *Let $\mathcal{X} \subset \mathbb{X}$ be finite and $r > 0$. The* (weighted) Čech complex *is the simplicial complex*

$$\check{C}ech_{\mathcal{X}}^{\mathbb{X}}(r) := \{\sigma \subset \mathcal{X} : \cap_{x \in \sigma} \mathbb{B}_{\mathbb{X}}(x, r) \neq \emptyset\}. \tag{7}$$

*The superscript $\mathbb{X}$ will be dropped when understood from the context.*

Another common choice is the *Vietoris-Rips complex*, also referred to as *Rips complex*, where simplexes are built based on pairwise distances among its vertices.

**Definition A.2 (Vietoris-Rips complex)** *Let $\mathcal{X} \subset \mathbb{X}$ be finite and $r > 0$. The* Vietoris-Rips complex $Rips_{\mathcal{X}}(r)$ *is the simplicial complex defined as*

$$Rips_{\mathcal{X}}(r) := \{\sigma \subset \mathcal{X} : d(x_i, x_j) < 2r, \forall x_i, x_j \in \sigma\}. \tag{8}$$

Note that from (7) and (8), the Čech complex and Vietoris-Rips complex have the following inter-leaving inclusion relationship

$$\check{C}ech_{\mathcal{X}}(r) \subset Rips_{\mathcal{X}}(r) \subset \check{C}ech_{\mathcal{X}}(2r).$$

In particular, when $\mathbb{X} \subset \mathbb{R}^d$ is a subset of a Euclidean space of dimension $d$, then the constant 2 can be tightened to $\sqrt{\frac{2d}{d+1}}$ (e.g., see Theorem 2.5 in de Silva and Ghrist [2007]):

$$\check{C}ech_{\mathcal{X}}(r) \subset Rips_{\mathcal{X}}(r) \subset \check{C}ech_{\mathcal{X}}\left(\sqrt{\frac{2d}{d+1}}r\right).$$

*Persistent homology* [Barannikov, 1994, Zomorodian and Carlsson, 2005, Edelsbrunner et al., 2000, Chazal et al., 2014a] is a multiscale approach to represent topological features of the complex $K$. A *filtration* $\mathcal{F}$ is a collection of subcomplexes approximating the data points at different resolutions, formally defined as follows.

**Definition A.3 (Filtration)** *A* filtration $\mathcal{F} = \{K_a \subset K\}_{a \in \mathbb{R}}$ *is a collection of subcomplexes of $K$ such that $a \leq b$ implies that $K_a \subset K_b$.*

For a filtration $\mathcal{F}$ and for each $k \in \mathbb{N}_0 = \mathbb{N} \cup \{0\}$, the associated persistent homology $PH_k\mathcal{F}$ is an ordered collection of $k$-th dimensional homologies, one for each element of $\mathcal{F}$.

**Definition A.4 (Persistent homology)** *Let $\mathcal{F}$ be a filtration and let $k \in \mathbb{N}_0$. The associated $k$-th* persistent homology $PH_k\mathcal{F}$ *is a collection of groups $\{H_k(K_a)\}_{a \in \mathbb{R}}$ of each subcomplex $K_a$ in $\mathcal{F}$ equipped with homomorphisms $\{\imath_k^{a,b}\}_{a \leq b}$, where $H_k(K_a)$ is the $k$-th dimensional homology group of $K_a$ and $\imath_k^{a,b} \colon H_k K_a \to H_k K_b$ is the homomorphism induced by the inclusion $K_a \subset K_b$.*

For the $k$-th persistent homology $PH_k\mathcal{F}$, the set of filtration levels at which a specific homology appears is always an interval $[b, d) \subset [-\infty, \infty]$, i.e. a specific homology is formed at some filtration value $b$ and dies when the inside hole is filled at another value $d > b$. To be more formally, the image of a specific homology class $\alpha$ in $H_k(K_a)$ is nonzero if and only if $b \leq a < d$. We often say that $\alpha$ is born at $b$ and dies at $d$. By considering these pairs as points in the plane, one obtains the *persistence diagram* as below.

**Definition A.5 (Persistence diagram)** *Let $\mathbb{R}_*^2 := \{(b, d) \in (\mathbb{R} \cup \infty)^2 : d > b\}$. Let $\mathcal{F}$ be a filtration and let $k \in \mathbb{N}_0$. The corresponding $k$-th persistence diagram $Dgm_k(\mathcal{F})$ is a finite multiset of $\mathbb{R}_*^2$, consisting of all pairs $(b, d)$, where $[b, d)$ is the interval of filtration values for which a specific homology class appears in $PH_k\mathcal{F}$. $b$ is called a birth time and $d$ is called a death time.*

When topological information of the underlying space is approximated by the observed points, it is often needed to compare two sets with respect to their metric structures. Here we present two distances on metric spaces, Hausdorff distance and Gromov-Hausdorff distance. We refer to Burago et al. [2001] for more details and other distances.

The *Hausdorff distance* [Burago et al., 2001, Definition 7.3.1] is on sets embedded in the same metric spaces. This distance measures how two sets are close to each other in the embedded metric space. When $S \subset \mathbb{X}$, we denote by $U_r(S)$ the $r$-neighborhood of a set $S$ in a metric space, i.e. $U_r(S) = \bigcup_{x \in S} \mathbb{B}_{\mathbb{X}}(x, r)$.

**Definition A.6 (Hausdorff distance)** *Let $\mathbb{X}$ be a metric space, and $X, Y \subset \mathbb{X}$ be a subset. The* Hausdorff distance *between $X$ and $Y$, denoted by $d_H(X, Y)$, is defined as*

$$d_H(X, Y) = \inf\{r > 0 : X \subset U_r(Y) \text{ and } Y \subset U_r(X)\}.$$

The *Gromov-Hausdorff distance* measures how two sets are far from being isometric to each other. To define the distance, we first define a relation between two sets called *correspondence*.

**Definition A.7** *Let $X$ and $Y$ be two sets. A* correspondence *between $X$ and $Y$ is a set $C \subset X \times Y$ whose projections to both $X$ and $Y$ are both surjective, i.e. for every $x \in X$, there exists $y \in Y$ such that $(x, y) \in C$, and for every $y \in Y$, there exists $x \in X$ with $(x, y) \in C$.*

For a correspondence, we define its *distortion* by how the metric structures of two sets differ by the correspondence.

**Definition A.8** *Let $X$ and $Y$ be two metric spaces, and $C$ be a correspondence between $X$ and $Y$. The* distortion *of $C$ is defined by*

$$dis(C) = \sup\left\{|d_X(x, x') - d_Y(y, y')| : (x, y), (x', y') \in C\right\}.$$

Now the Gromov-Hausdorff distance [Burago et al., 2001, Theorem 7.3.25] is defined as the smallest possible distortion between two sets.

**Definition A.9 (Gromov-Hausdorff distance)** *Let $X$ and $Y$ be two metric spaces. The* Gromov-Hausdorff distance *between $X$ and $Y$, denoted as $d_{GH}(X, Y)$, is defined as*

$$d_{GH}(X, Y) = \frac{1}{2} \inf_C dis(C),$$

*where the infimum is over all correspondences between $X$ and $Y$.*

# B  Bottleneck distance and Wasserstein distance

Our stability bound in Theorem 4.1 is based on the bottleneck distance, while the stability bound in Hofer et al. [2017] is based on Wasserstein distance. Hence to compare these bounds, we need to understand the relationship between the bottleneck distance and Wasserstein distance. We already know that the Wasserstein distance is lower bounded by the bottleneck distance. Here, we will find a tighter lower bound for the ratio of the Wasserstein distance to the bottleneck distance.

Before analyzing the relationship between them, we first show a claim.

**Claim B.1** *Let $\mathcal{D}, \mathcal{D}'$ be two persistence diagrams. For $t > 0$, let $n_t \in \mathbb{N}$ be satisfying the followings: for any two diagrams $\mathcal{D}_t, \mathcal{D}'_t$ with $d_B(\mathcal{D}, \mathcal{D}_t) \leq t$ and $d_B(\mathcal{D}', \mathcal{D}'_t) \leq t$, either $|\mathcal{D}_t \backslash \mathcal{D}'_t| \geq n_t$ or $|\mathcal{D}'_t \backslash \mathcal{D}_t| \geq n_t$ holds. Then for any bijection $\gamma : \bar{\mathcal{D}} \to \bar{\mathcal{D}}'$, the number of paired points with being at least $2t$ apart in $L_\infty$ distance is greater or equal to $n_t$, i.e.,*

$$\left| \left\{ p \in \bar{\mathcal{D}} : \| p - \gamma(p) \|_\infty > 2t \right\} \right| \geq n_t.$$

And then, we get a lower bound for the ratio of Wasserstein distance to the bottleneck distance.

**Proposition B.1** *Let $\mathcal{D}, \mathcal{D}'$ be two persistence diagrams. For $t > 0$, let $n_t \in \mathbb{N}$ be satisfying the followings: for any two diagrams $\mathcal{D}_t, \mathcal{D}'_t$ with $d_B(\mathcal{D}, \mathcal{D}_t) \leq t$ and $d_B(\mathcal{D}', \mathcal{D}'_t) \leq t$, either $|\mathcal{D}_t \backslash \mathcal{D}'_t| \geq n_t$ or $|\mathcal{D}'_t \backslash \mathcal{D}_t| \geq n_t$ holds. Then, the ratio of q-Wasserstein distance to the bottleneck distance is bounded as*

$$\frac{W_q(\mathcal{D}, \mathcal{D}')}{d_B(\mathcal{D}, \mathcal{D}')} \geq \left( 1 + \left( \frac{2t}{d_B(\mathcal{D}, \mathcal{D}')} \right)^q (n_t - 1) \right)^{\frac{1}{q}}.$$

## C   Stability for Vietoris-Rips and Cech filtration

When we use Vietoris-Rips or Čech filtration, our result can be turned into the stability result with respect to points in Euclidean space. Let $\mathbb{X}, \mathbb{Y} \subset \mathbb{R}^d$ be two bounded sets. The next corollary re-states our stability theorem with respect to points in $\mathbb{R}^d$.

**Corollary C.1** *Let $X, Y$ be any $\epsilon$-coverings of $\mathbb{X}, \mathbb{Y}$, and let $\mathcal{D}_X, \mathcal{D}_Y$ denote persistence diagrams induced from the Vietoris-Rips or Čech filtration on $X, Y$ respectively. Then we have*

$$|S_{\boldsymbol{\theta}, \boldsymbol{\omega}}(\mathcal{D}_X; \nu) - S_{\boldsymbol{\theta}, \boldsymbol{\omega}}(\mathcal{D}_Y; \nu)| \leq 2L_g \left( d_{GH}(\mathbb{X}, \mathbb{Y}) + 2\epsilon \right). \tag{9}$$

The proof is given in Appendix E.7. Corollary C.1 implies that if we assume our observed data points are sufficiently decent quality in the sense that $\epsilon \to 0$, then our topological layers constructed on those observed points are stable with respect to small perturbations of the true representation under proper persistent homologies. Here, $\epsilon$ could be interpreted as uncertainty from incomplete sampling. This means the topological information embedded in the proposed layer is robust against small sampling noise or data corruption by missingness.

Moreover, since Gromov-Hausdorff distance is upper bounded by Hausdorff distance, the result in Corollary C.1 also holds when we use $d_H(X, Y)$ in place of $d_{GH}(X, Y)$ in RHS of (9).

**Remark 2** *In fact, when we have very dense data that have been well-sampled uniformly over the true representation so that $\epsilon \to 0$, our result in (9) converges to the following:*

$$|S_{\boldsymbol{\theta}, \boldsymbol{\omega}}(\mathcal{D}_{\mathbb{X}}; \nu) - S_{\boldsymbol{\theta}, \boldsymbol{\omega}}(\mathcal{D}_{\mathbb{Y}}; \nu)| \leq 2L_g d_{GH}(\mathbb{X}, \mathbb{Y}).$$

## D   Differentiability of DTM function

Here we provide a specific example of computing $\frac{\partial f(\varsigma)}{\partial X_j}$ when $f$ is the DTM filtration which has not been explored in previous approaches. We first consider the case of (4) where $X_j$'s are data points, as in Proposition D.1. See Appendix E.8 for the proof.

**Proposition D.1** *When $X_j$'s and $\varsigma$ satisfy that $\sum_{X_i \in N_k(y)} \varpi_i \| X_i - y_l \|^r$ are different for each $y_l \in \varsigma$, then $f(\varsigma)$ is differentiable with respect to $X_j$ and*

$$\frac{\partial f(\varsigma)}{\partial X_j} = \frac{\varpi'_j \| X_j - y \|^{r-2} (X_j - y) I(X_j \in N_k(y))}{\left( \hat{d}_{m_0}(y) \right)^{r-1} m_0 \sum_{i=1}^n \varpi_i},$$

*where $I$ is an indicator function and $y = \arg\max_{z \in \varsigma} \hat{d}_{m_0}(z)$. In particular, $f$ is differentiable a.e. with respect to Lebesgue measure on $X$.*

Similarly, we consider the case of (5) where $X_j$'s are weights, as in Proposition D.2. See Appendix E.9 for the proof.

**Proposition D.2** *When $X_j$'s and $\varsigma$ satisfy that $\sum_{Y_i \in N_k(y)} X_i' \|Y_i - y_l\|^r$ are different for each $y_l \in \varsigma$, then $f(\varsigma)$ is differentiable with respect to $X_j$ and*

$$\frac{\partial f(\varsigma)}{\partial X_j} = \frac{\|Y_j - y\|^r I(Y_j \in N_k(y)) - m_0 \left(\hat{d}_{m_0}(y)\right)^r}{r \left(\hat{d}_{m_0}(y)\right)^{r-1} m_0 \sum_{i=1}^n X_i},$$

*where $y = \arg\max_{y \in \varsigma_i} \hat{d}_{m_0}(y)$. In particular, $f$ is differentiable a.e. with respect to Lebesgue measure on $X$ and $Y$.*

Computation of $\frac{\partial h_{\text{top}}}{\partial \boldsymbol{\mu}_i}$, $\frac{\partial h_{\text{top}}}{\partial \varsigma_i}$ are simpler and can be done in a similar fashion. In the experiments, we set $r = 2$.

## E Proofs

### E.1 Proof of Theorem 3.1

For computing $\frac{\partial h_{\text{top}}}{\partial X_j}$, note that it can be expanded using the chain role as

$$\frac{\partial h_{\text{top}}}{\partial X_j} = \sum_i \frac{\partial h_{\text{top}}}{\partial b_i} \frac{\partial b_i}{\partial X_j} + \sum_i \frac{\partial h_{\text{top}}}{\partial d_i} \frac{\partial d_i}{\partial X_j}, \tag{10}$$

and hence we need to compute $\frac{\partial \mathcal{D}_X}{\partial X} = \left\{ \left( \frac{\partial b_i}{\partial X_j}, \frac{\partial d_i}{\partial X_j} \right) \right\}_{(b_i, d_i) \in \mathcal{D}_X, X_j \in X}$ and $\frac{\partial h_{\text{top}}}{\partial \mathcal{D}_X} = \left\{ \left( \frac{\partial h_{\text{top}}}{\partial b_i}, \frac{\partial h_{\text{top}}}{\partial d_i} \right) \right\}_{(b_i, d_i) \in \mathcal{D}_X}$ to compute $\frac{\partial h_{\text{top}}}{\partial X_j}$.

We first compute $\frac{\partial \mathcal{D}_X}{\partial X}$. Let $K$ be the simplicial complex, and suppose all the simplices are ordered in the filtration so that the values of $f$ are nondecreasing, i.e. if $\varsigma$ comes earlier than $\tau$ then $f(\varsigma) \leq f(\tau)$. Note that the map $\xi$ from each birth-death point $(b_i, d_i) \in \mathcal{D}_X$ to a pair of simplices $(\beta_i, \delta_i)$ is simply the pairing returned by the standard persistence diagram [Carlsson et al., 2005]. Let $\gamma$ be the homological feature corresponding to $(b_i, d_i)$, then the birth simplex $\beta_i$ is the simplex that forms $\gamma$ in $K_{b_i} = f^{-1}(-\infty, b_i]$, and the death simplex $\delta_i$ is the simplex that causes $\gamma$ to collapse in $K_{d_i} = f^{-1}(-\infty, d_i]$. For example, if $\gamma$ were to be a 1-dimensional feature, then $\beta_i$ is the edge in $K_{b_i}$ that forms the loop corresponding to $\gamma$, and $\delta_i$ is the triangle in $K_{d_i}$ which incurs the loop corresponding to $\gamma$ can be contracted in $K_{d_i}$.

Now, $f(\xi(b_i)) = f(\beta_i) = b_i$ and $f(\xi(d_i)) = f(\delta_i) = d_i$, and from $\xi$ being locally constant on $X$,

$$\frac{\partial b_i}{\partial X_j} = \frac{\partial f(\xi(b_i))}{\partial X_j} = \frac{\partial f(\beta_i)}{\partial X_j}, \quad \frac{\partial d_i}{\partial X_j} = \frac{\partial f(\xi(d_i))}{\partial X_j} = \frac{\partial f(\delta_i)}{\partial X_j}. \tag{11}$$

Therefore, the derivatives of the birth value and the death value are the derivatives of the filtration function evaluated at the corresponding pair of simplices. And $\frac{\partial \mathcal{D}_X}{\partial X} = \left\{ \left( \frac{\partial b_i}{\partial X_j}, \frac{\partial d_i}{\partial X_j} \right) \right\}_{(b_i, d_i) \in \mathcal{D}_X, X_j \in X}$ is the collection of these derivatives, hence applying (11) gives

$$\frac{\partial \mathcal{D}_X}{\partial X} = \left\{ \left( \frac{\partial b_i}{\partial X_j}, \frac{\partial d_i}{\partial X_j} \right) \right\}_{(b_i, d_i) \in \mathcal{D}_X, X_j \in X} = \left\{ \left( \frac{\partial f(\beta_i)}{\partial X_j}, \frac{\partial f(\delta_i)}{\partial X_j} \right) \right\}_{\xi^{-1}(\beta_i, \delta_i) \in \mathcal{D}_X, X_j \in X}. \tag{12}$$

Now, we compute $\frac{\partial h_{\text{top}}}{\partial \mathcal{D}_X} = \left\{ \left( \frac{\partial h_{\text{top}}}{\partial b_i}, \frac{\partial h_{\text{top}}}{\partial d_i} \right) \right\}_{(b_i, d_i) \in \mathcal{D}_X}$. Computing $\frac{\partial h_{\text{top}}}{\partial b_i}$ can be done by applying the chain role on $h_{\text{top}} = S_{\boldsymbol{\theta}, \boldsymbol{\omega}} = g_{\boldsymbol{\theta}} \circ \overline{\boldsymbol{\Lambda}}_{\boldsymbol{\omega}}$ as

$$\frac{\partial h_{\text{top}}}{\partial b_i} = \frac{\partial S_{\boldsymbol{\theta}, \boldsymbol{\omega}}}{\partial b_i} = \frac{\partial (g_{\boldsymbol{\theta}} \circ \overline{\boldsymbol{\Lambda}}_{\boldsymbol{\omega}})}{\partial b_i} = \nabla g_{\boldsymbol{\theta}} \circ \frac{\partial \overline{\boldsymbol{\Lambda}}_{\boldsymbol{\omega}}}{\partial b_i} = \sum_{l=1}^m \frac{\partial g_{\boldsymbol{\theta}}}{\partial x_l} \frac{\partial \overline{\lambda}_{\boldsymbol{\omega}}(l\nu)}{\partial b_i}, \tag{13}$$

where we use $x_l$ as the shorthand notation for the input of the function $g_{\boldsymbol{\theta}}$. Then, applying $\overline{\lambda}_{\boldsymbol{\omega}}(l\nu) = \sum_{k=1}^{K_{max}} \omega_k \lambda_k(l\nu)$ to (13) gives

$$\frac{\partial h_{\text{top}}}{\partial b_i} = \sum_{l=1}^m \frac{\partial g_{\boldsymbol{\theta}}}{\partial x_l} \sum_{k=1}^{K_{\max}} \omega_k \frac{\partial \lambda_k(l\nu)}{\partial b_i}. \tag{14}$$

Similarly, $\frac{\partial h_{\text{top}}}{\partial d_i}$ can be computed as

$$\frac{\partial h_{\text{top}}}{\partial d_i} = \sum_{l=1}^{m} \frac{\partial g_{\boldsymbol{\theta}}}{\partial x_l} \sum_{k=1}^{K_{\max}} \omega_k \frac{\partial \lambda_k(lv)}{\partial d_i}. \tag{15}$$

And therefore, $\frac{\partial h_{\text{top}}}{\partial \mathcal{D}_X}$ is the collection of these derivatives from (14) and (15), i.e.,

$$\frac{\partial h_{\text{top}}}{\partial \mathcal{D}_X} = \left\{ \left( \sum_{l=1}^{m} \frac{\partial g_{\boldsymbol{\theta}}}{\partial x_l} \sum_{k=1}^{K_{\max}} \omega_k \frac{\partial \lambda_k(lv)}{\partial b_i}, \sum_{l=1}^{m} \frac{\partial g_{\boldsymbol{\theta}}}{\partial x_l} \sum_{k=1}^{K_{\max}} \omega_k \frac{\partial \lambda_k(lv)}{\partial d_i} \right) \right\}_{(b_i, d_i) \in \mathcal{D}_X}. \tag{16}$$

Hence, $\frac{\partial h_{\text{top}}}{\partial X}$ can be computed by applying (12) and (16) to (10) as

$$\begin{aligned}
\frac{\partial h_{\text{top}}}{\partial X_j} &= \sum_i \frac{\partial h_{\text{top}}}{\partial b_i} \frac{\partial b_i}{\partial X_j} + \sum_i \frac{\partial h_{\text{top}}}{\partial d_i} \frac{\partial d_i}{\partial X_j} \\
&= \sum_i \frac{\partial f(\beta_i)}{\partial X_j} \sum_{l=1}^{m} \frac{\partial g_{\boldsymbol{\theta}}}{\partial x_l} \sum_{k=1}^{K_{\max}} \omega_k \frac{\partial \lambda_k(lv)}{\partial b_i} + \sum_i \frac{\partial f(\delta_i)}{\partial X_j} \sum_{l=1}^{m} \frac{\partial g_{\boldsymbol{\theta}}}{\partial x_l} \sum_{k=1}^{K_{\max}} \omega_k \frac{\partial \lambda_k(lv)}{\partial d_i}.
\end{aligned}$$

### E.2 Proof of Theorem 4.1

Let $\mathcal{D}$ and $\mathcal{D}'$ be two persistence diagrams and let $\lambda$ and $\lambda'$ be their persistence landscapes. All the quantities derived from $\mathcal{D}'$ are denoted by a variable name with the superscript $\prime$ hereafter (e.g., $\lambda'_k(t), \overline{\boldsymbol{\Lambda}'}_{\boldsymbol{\omega}}$).

For the stability of the structure element $S_{\boldsymbol{\theta},\boldsymbol{\omega}}$, we first expand the difference between $S_{\boldsymbol{\theta},\boldsymbol{\omega}}(\mathcal{D};\nu)$ and $S_{\boldsymbol{\theta},\boldsymbol{\omega}}(\mathcal{D}';\nu)$ using $S_{\boldsymbol{\theta},\boldsymbol{\omega}} = g_{\boldsymbol{\theta}} \circ \overline{\boldsymbol{\Lambda}}_{\boldsymbol{\omega}}$ as

$$|S_{\boldsymbol{\theta},\boldsymbol{\omega}}(\mathcal{D};\nu) - S_{\boldsymbol{\theta},\boldsymbol{\omega}}(\mathcal{D}';\nu)| = \left| g_{\boldsymbol{\theta}}\left(\overline{\boldsymbol{\Lambda}}_{\boldsymbol{\omega}}\right) - g_{\boldsymbol{\theta}}\left(\overline{\boldsymbol{\Lambda}'}_{\boldsymbol{\omega}}\right) \right|. \tag{17}$$

Then, RHS of (17) is bounded by applying the Lipschitz condition of the function $g_{\boldsymbol{\theta}}$ as

$$\left| g_{\boldsymbol{\theta}}\left(\overline{\boldsymbol{\Lambda}}_{\boldsymbol{\omega}}\right) - g_{\boldsymbol{\theta}}\left(\overline{\boldsymbol{\Lambda}'}_{\boldsymbol{\omega}}\right) \right| \leq L_g \left\| \overline{\boldsymbol{\Lambda}}_{\boldsymbol{\omega}} - \overline{\boldsymbol{\Lambda}'}_{\boldsymbol{\omega}} \right\|_{\infty}. \tag{18}$$

Then for $\left\| \overline{\boldsymbol{\Lambda}}_{\boldsymbol{\omega}} - \overline{\boldsymbol{\Lambda}'}_{\boldsymbol{\omega}} \right\|_{\infty}$, note that $\overline{\boldsymbol{\Lambda}}_{\boldsymbol{\omega}}, \overline{\boldsymbol{\Lambda}'}_{\boldsymbol{\omega}} \in \mathbb{R}^m$, the $L_{\infty}$ difference of $\overline{\boldsymbol{\Lambda}}_{\boldsymbol{\omega}}$ and $\overline{\boldsymbol{\Lambda}'}_{\boldsymbol{\omega}}$ is bounded as

$$\begin{aligned}
\left\| \overline{\boldsymbol{\Lambda}}_{\boldsymbol{\omega}} - \overline{\boldsymbol{\Lambda}'}_{\boldsymbol{\omega}} \right\|_{\infty} &= \max_{0 \leq i \leq m-1} \left| \overline{\lambda}_{\boldsymbol{\omega}}(T_{\min} + i\nu) - \overline{\lambda}'_{\boldsymbol{\omega}}(T_{\min} + i\nu) \right| \\
&\leq \sup_{t \in [0,T]} \left| \overline{\lambda}_{\boldsymbol{\omega}}(t) - \overline{\lambda}'_{\boldsymbol{\omega}}(t) \right| = m^{1/2} \left\| \overline{\lambda}_{\boldsymbol{\omega}} - \overline{\lambda}'_{\boldsymbol{\omega}} \right\|_{\infty}.
\end{aligned} \tag{19}$$

Now, for bounding $\left\| \overline{\lambda}_{\boldsymbol{\omega}} - \overline{\lambda}'_{\boldsymbol{\omega}} \right\|_{\infty}$, we first consider the pointwise difference $|\overline{\lambda}_{\boldsymbol{\omega}}(t) - \overline{\lambda}'_{\boldsymbol{\omega}}(t)|$. For all $t \in [0,T]$, the difference between $\overline{\lambda}_{\boldsymbol{\omega}}(t)$ and $\overline{\lambda}'_{\boldsymbol{\omega}}(t)$ is bounded as

$$\begin{aligned}
\left| \overline{\lambda}_{\boldsymbol{\omega}}(t) - \overline{\lambda}'_{\boldsymbol{\omega}}(t) \right| &= \left| \frac{1}{\sum_k \omega_k} \sum_{k=1}^{K_{\max}} \omega_k \lambda_k(t) - \frac{1}{\sum_k \omega_k} \sum_{k=1}^{K_{\max}} \omega_k \lambda'_k(t) \right| \\
&\leq \frac{1}{\sum_k \omega_k} \sum_{k=1}^{K_{\max}} \omega_k |\lambda_k(t) - \lambda'_k(t)| \\
&\leq \sup_{1 \leq k \leq K_{\max}, t \in [0,T]} |\lambda_k(t) - \lambda'_k(t)| = \max_{1 \leq k \leq K_{\max}} \|\lambda_k - \lambda'_k\|_{\infty}.
\end{aligned} \tag{20}$$

And hence $\left\| \overline{\lambda}_{\boldsymbol{\omega}} - \overline{\lambda}'_{\boldsymbol{\omega}} \right\|_{\infty}$ is bounded by $\max_{1 \leq k \leq K_{\max}} \|\lambda_k - \lambda'_k\|_{\infty}$ as well, i.e.,

$$\left\| \overline{\lambda}_{\boldsymbol{\omega}} - \overline{\lambda}'_{\boldsymbol{\omega}} \right\|_{\infty} = \sup_{t \in [0,T]} \left| \overline{\lambda}_{\boldsymbol{\omega}}(t) - \overline{\lambda}'_{\boldsymbol{\omega}}(t) \right| \leq \max_{1 \leq k \leq K_{\max}} \|\lambda_k - \lambda'_k\|_{\infty}. \tag{21}$$

Then for all $k = 1, \ldots, K_{\max}$, the $\infty$-landscape distance $\|\lambda_k - \lambda'_k\|_\infty$ is bounded by the bottleneck distance $d_B(\mathcal{D}, \mathcal{D}')$ from Theorem 13 in Bubenik [2015], i.e.

$$\|\lambda_k - \lambda'_k\|_\infty \leq d_B(\mathcal{D}, \mathcal{D}').  \tag{22}$$

Hence, applying (18), (19), (21), (22) to (17) gives the stated stability result as

$$
\begin{aligned}
|S_{\boldsymbol{\theta},\boldsymbol{\omega}}(\mathcal{D}; \nu) - S_{\boldsymbol{\theta},\boldsymbol{\omega}}(\mathcal{D}'; \nu)| = \left| g_{\boldsymbol{\theta}}\left(\overline{\boldsymbol{\Lambda}}_{\boldsymbol{\omega}}\right) - g_{\boldsymbol{\theta}}\left(\overline{\boldsymbol{\Lambda}'}_{\boldsymbol{\omega}}\right) \right| &\leq L_g \left\| \overline{\boldsymbol{\Lambda}}_{\boldsymbol{\omega}} - \overline{\boldsymbol{\Lambda}'}_{\boldsymbol{\omega}} \right\|_\infty \\
&\leq L_g \left\| \overline{\lambda}_{\boldsymbol{\omega}} - \overline{\lambda}'_{\boldsymbol{\omega}} \right\|_\infty \leq L_g \max_{1 \leq k \leq K_{\max}} \|\lambda_k - \lambda'_k\|_\infty \\
&\leq L_g d_B(\mathcal{D}, \mathcal{D}').
\end{aligned}
$$

### E.3 Proof of Corollary 4.1

First note that the result of Hofer et al. [2017] used $W_1$ Wasserstein distance with $L_r$ norm for $\forall r \in \mathbb{N}$, which will be denoted by $W_1^{L_r}$ in this proof. That is,

$$W_1^{L_r}(\mathcal{D}, \mathcal{D}') \coloneqq \inf_\gamma \sum_{p \in \mathcal{D}_X} \|p - \gamma(p)\|_r$$

where $\gamma$ ranges over all bijections $\mathcal{D} \to \mathcal{D}'$ (i.e., $W_1^{L_\infty}$ corresponds to $W_1$ in our definition 2.2). Then, $\|\cdot\|_r \geq \|\cdot\|_\infty$ implies that $W_1^{L_r}$ is lower bounded by $W_1$, i.e.

$$W_1^{L_r}(\mathcal{D}, \mathcal{D}') \geq W_1(\mathcal{D}, \mathcal{D}').  \tag{23}$$

Now, let $c_K$ denote the Lipschitz constant in Hofer et al. [2017, Theorem 1] and $c_{g_\theta}$ denote the constant term in our result in Theorem 4.1, i.e. $c_{g_\theta} = L_g \left(\frac{T}{\nu}\right)^{1/2}$. We want to upper bound the ratio $\frac{c_{g_\theta} d_B(\mathcal{D}, \mathcal{D}')}{c_K W_1^{L_r}(\mathcal{D}, \mathcal{D}')}$. This directly comes from (23) and Proposition B.1 as

$$\frac{c_{g_\theta} d_B(\mathcal{D}, \mathcal{D}')}{c_K W_1^{L_r}(\mathcal{D}, \mathcal{D}')} \geq \frac{c_{g_\theta}}{c_K} \frac{d_B(\mathcal{D}, \mathcal{D}')}{W_1(\mathcal{D}, \mathcal{D}')} \geq \frac{c_{g_\theta}}{c_K} \frac{1}{1 + \frac{2t}{d_B(\mathcal{D}, \mathcal{D}')}(n_t - 1)}.$$

Finally, we define $C_{g_\theta, T, \nu} \coloneqq \frac{c_{g_\theta, T, \nu}}{c_K}$, and the result follows.

It should be noted that the bound is actually very loose. However, we can still conclude that our bound is tighter than that of Hofer et al. [2017] at polynomial rates.

### E.4 Proof of Theorem 4.2

We first bound the difference between $S_{\boldsymbol{\theta},\boldsymbol{\omega}}(\mathcal{D}_X; \nu)$ and $S_{\boldsymbol{\theta},\boldsymbol{\omega}}(\mathcal{D}_P; \nu)$ using Theorem 4.1 as

$$|S_{\boldsymbol{\theta},\boldsymbol{\omega}}(\mathcal{D}_X; \nu) - S_{\boldsymbol{\theta},\boldsymbol{\omega}}(\mathcal{D}_P; \nu)| \leq L_g d_B(\mathcal{D}_X, \mathcal{D}_P).  \tag{24}$$

It is left to further bound the bottleneck distance $d_B(\mathcal{D}_X, \mathcal{D}_P)$. The bottleneck distance between two diagrams $\mathcal{D}_X$ and $\mathcal{D}_P$ is bounded by the stability theorem of persistent homology as

$$d_B(\mathcal{D}_X, \mathcal{D}_P) \leq \|d_{P_n, m_0} - d_{P, m_0}\|_\infty.  \tag{25}$$

Then, from $r = 2$ in the DTM function, the $L_\infty$ distance between $d_{P_n, m_0}$ and $d_{P, m_0}$ is bounded by the stability of DTM function (Theorem 3.5 from Chazal et al. [2011]) as

$$\|d_{P_n, m_0} - d_{P, m_0}\|_\infty \leq m_0^{-1/2} W_2(P_n, P).  \tag{26}$$

Hence, combining (24), (25), and (26) altogether gives the stated stability result as

$$|S_{\boldsymbol{\theta},\boldsymbol{\omega}}(\mathcal{D}_X; \nu) - S_{\boldsymbol{\theta},\boldsymbol{\omega}}(\mathcal{D}_P; \nu)| \leq L_g m_0^{-1/2} W_2(P_n, P).$$

### E.5 Proof of Claim B.1

Let $\gamma\colon \mathcal{D} \to \mathcal{D}'$ be any bijection and let $\mathcal{S} := \big\{p \in \bar{\mathcal{D}} : \|p - \gamma(p)\|_{\infty} > 2t\big\}$. Then for $p \in \bar{\mathcal{D}}$ with $\|p - \gamma(p)\|_{\infty} \leq 2t$, there exists $\beta(p) \in \mathbb{R}_{*}^{2}$ such that $\|p - \beta(p)\|_{\infty} \leq t$ and $\|\beta(p) - \gamma(p)\|_{\infty} \leq t$. Now, define two diagrams $\mathcal{D}_t, \mathcal{D}_t'$ as follows:

$$\mathcal{D}_t = \mathcal{S} \cup \big\{\beta(p) : p \in \bar{\mathcal{D}}\backslash\mathcal{S}\big\} \backslash Diag,$$
$$\mathcal{D}_t' = \mathcal{S}' \cup \big\{\beta(p) : p \in \bar{\mathcal{D}}\backslash\mathcal{S}\big\} \backslash Diag,$$

where $\mathcal{S}' := \big\{\gamma(p) : p \in \bar{\mathcal{D}}\big\}$. Then, $d_B(\mathcal{D}, \mathcal{D}_t) \leq t$ and $d_B(\mathcal{D}', \mathcal{D}_t') \leq t$ from the construction. Hence from the definition of $n_t$, either $|\mathcal{D}_t \backslash \mathcal{D}_t'| \geq n_t$ or $|\mathcal{D}_t' \backslash \mathcal{D}_t| \geq n_t$ holds. Now, note that

$$\mathcal{D}_t \backslash \mathcal{D}_t' \subset S \qquad \text{and} \qquad \mathcal{D}_t' \backslash \mathcal{D}_t \subset \mathcal{S}'.$$

And $|S| = |S'|$, and hence we get the claimed result as

$$|S| \geq n_t.$$

### E.6 Proof of Proposition B.1

We consider a bijection $\gamma^*$ that realizes the $q$-Wasserstein distance between $\mathcal{D}$ and $\mathcal{D}'$: i.e. $\gamma^* = \operatorname*{arginf}_{\gamma} \sum_{p \in \mathcal{D}} \|p - \gamma(p)\|_{\infty}^{q}$. Then we have that

$$d_B(\mathcal{D}, \mathcal{D}')^q \leq \sup_{p \in \mathcal{D}} \|p - \gamma^*(p)\|_{\infty}^{q}. \tag{27}$$

On the other hand, if we let $p^* = \operatorname*{argsup}_{p \in \mathcal{D}} \|p - \gamma^*(p)\|_{\infty}$, we have

$$W_q(\mathcal{D}, \mathcal{D}')^q = \sum_{p \in \mathcal{D}} \|p - \gamma^*(p)\|_{\infty}^{q} = \sup_{p \in \mathcal{D}} \|p - \gamma^*(p)\|_{\infty}^{q} + \sum_{p \neq p^*} \|p - \gamma^*(p)\|_{\infty}^{q}.$$

Note that from Claim B.1, $\big|\big\{p \in \bar{\mathcal{D}} : \|p - \gamma^*(p)\|_{\infty} > 2t\big\}\big| \geq n_t$. And hence $W_q(\mathcal{D}, \mathcal{D}')^q$ can be lower bounded as

$$W_q(\mathcal{D}, \mathcal{D}')^q = \sup_{p \in \mathcal{D}} \|p - \gamma^*(p)\|_{\infty}^{q} + \sum_{p \neq p^*} \|p - \gamma^*(p)\|_{\infty}^{q} \tag{28}$$

$$\geq \sup_{p \in \mathcal{D}} \|p - \gamma^*(p)\|_{\infty}^{q} + (2t)^q (n_t - 1). \tag{29}$$

Now, we lower bound the ratio $\frac{W_q(\mathcal{D}, \mathcal{D}')^q}{d_B(\mathcal{D}, \mathcal{D}')^q}$. By (27) and (29), this can be done as follows.

$$\frac{W_q(\mathcal{D}, \mathcal{D}')^q}{d_B(\mathcal{D}, \mathcal{D}')^q} \geq \frac{\sup_{p \in \mathcal{D}} \|p - \gamma^*(p)\|_{\infty}^{q} + (2t)^q (n_t - 1)}{d_B(\mathcal{D}, \mathcal{D}')^q}$$

$$\geq 1 + \left(\frac{2t}{d_B(\mathcal{D}, \mathcal{D}')}\right)^q (n_t - 1).$$

And hence the ratio of the Wasserstein distance to thw bottleneck distance $\frac{W_q(\mathcal{D}, \mathcal{D}')}{d_B(\mathcal{D}, \mathcal{D}')}$ is correspondingly lower bounded as

$$\frac{W_q(\mathcal{D}, \mathcal{D}')}{d_B(\mathcal{D}, \mathcal{D}')} \geq \left(\frac{W_q(\mathcal{D}, \mathcal{D}')^q}{d_B(\mathcal{D}, \mathcal{D}')^q}\right)^{\frac{1}{q}} \geq \left(1 + \left(\frac{2t}{d_B(\mathcal{D}, \mathcal{D}')}\right)^q (n_t - 1)\right)^{\frac{1}{q}}.$$

### E.7 Proof of Corollary C.1

The difference between $S_{\boldsymbol{\theta},\boldsymbol{\omega}}(\mathcal{D}_X; \nu)$ and $S_{\boldsymbol{\theta},\boldsymbol{\omega}}(\mathcal{D}_Y; \nu)$ is bounded by Theorem 4.1 as

$$|S_{\boldsymbol{\theta},\boldsymbol{\omega}}(\mathcal{D}_X; \nu) - S_{\boldsymbol{\theta},\boldsymbol{\omega}}(\mathcal{D}_Y; \nu)| \leq L_g d_B(\mathcal{D}_X, \mathcal{D}_Y), \tag{30}$$

hence it suffices to show

$$d_B(\mathcal{D}_X, \mathcal{D}_Y) < 2(d_{GH}(\mathbb{X}, \mathbb{Y}) + 2\epsilon). \tag{31}$$

To show (31), we first apply the triangle inequality as

$$d_B\left(\mathcal{D}_X, \mathcal{D}_Y\right) \le d_B\left(\mathcal{D}_X, \mathcal{D}_\mathbb{X}\right) + d_B\left(\mathcal{D}_\mathbb{X}, \mathcal{D}_\mathbb{Y}\right) + d_B\left(\mathcal{D}_\mathbb{Y}, \mathcal{D}_Y\right). \tag{32}$$

And note that since $\mathbb{X}, \mathbb{Y}, X, Y$ are all bounded in Euclidean space, they are totally bounded metric spaces. Thus by Theorem 5.2 in Chazal et al. [2014a], the bottleneck distance between any two diagrams is bounded by Gromov-Hausdorff distance, and in particular,

$$d_B\left(\mathcal{D}_\mathbb{X}, \mathcal{D}_\mathbb{Y}\right) \le 2 d_{GH}\left(\mathbb{X}, \mathbb{Y}\right),$$
$$d_B\left(\mathcal{D}_X, \mathcal{D}_\mathbb{X}\right) \le 2 d_{GH}\left(X, \mathbb{X}\right), \quad d_B\left(\mathcal{D}_\mathbb{Y}, \mathcal{D}_Y\right) \le 2 d_{GH}\left(\mathbb{Y}, Y\right). \tag{33}$$

And then since the Gromov-Hausdorff distance is bounded by the Hausdorff distance,

$$d_{GH}\left(X, \mathbb{X}\right) \le d_H\left(X, \mathbb{X}\right), \quad d_{GH}\left(\mathbb{Y}, Y\right) \le d_H\left(\mathbb{Y}, Y\right). \tag{34}$$

And the Hausdorff distance between $X$ and $\mathbb{X}$ or $Y$ and $\mathbb{Y}$ is bounded by $\epsilon$ by the assumption that $X, Y$ are $\epsilon$-coverings of $\mathbb{X}, \mathbb{Y}$, respectively, i.e.,

$$d_H\left(X, \mathbb{X}\right) < \epsilon, \quad d_H\left(\mathbb{Y}, Y\right) < \epsilon. \tag{35}$$

Hence combining (32), (33), (34), and (35) gives (31) as

$$\begin{aligned}
d_B\left(\mathcal{D}_X, \mathcal{D}_Y\right) &\le d_B\left(\mathcal{D}_X, \mathcal{D}_\mathbb{X}\right) + d_B\left(\mathcal{D}_\mathbb{X}, \mathcal{D}_\mathbb{Y}\right) + d_B\left(\mathcal{D}_\mathbb{Y}, \mathcal{D}_Y\right) \\
&\le 2\left(d_{GH}\left(X, \mathbb{X}\right) + d_{GH}\left(\mathbb{X}, \mathbb{Y}\right) + d_{GH}\left(\mathbb{Y}, Y\right)\right) \\
&\le 2\left(d_H\left(X, \mathbb{X}\right) + d_{GH}\left(\mathbb{X}, \mathbb{Y}\right) + d_H\left(\mathbb{Y}, Y\right)\right) \\
&< 2\left(d_{GH}\left(\mathbb{X}, \mathbb{Y}\right) + 2\epsilon\right).
\end{aligned}$$

Now, the results follows from (30) and (31).

### E.8 Proof of Proposition D.1

From (4), note that for any $y \in \varsigma$, $\hat{d}_{m_0}(y)$ is expanded as

$$\hat{d}_{m_0}(y) = \left(\frac{\sum_{X_i \in N_k(y)} \varpi_i' \left\|X_i - y\right\|^r}{m_0 \sum_{i=1}^n \varpi_i}\right)^{1/r}, \tag{36}$$

where $k$ is such that $\sum_{X_i \in N_{k-1}(y)} \varpi_i < m_0 \sum_{i=1}^n \varpi_i \le \sum_{X_i \in N_k(y)} \varpi_i$, and $\varpi_i' = \sum_{X_j \in N_k(y)} \varpi_j - m_0 \sum_{j=1}^n \varpi_j$ for one of $X_i$'s that is $k$-th nearest neighbor of $y$ and $\omega_i' = \omega_i$ otherwise. Hence, by letting $y = \arg\max_{z \in \varsigma} \hat{d}_{m_0}(z)$ applying to (36), the filtration function $f_X$ at simplex $\varsigma$ becomes

$$f_X(\varsigma) = \hat{d}_{X,m_0}(y) = \left(\frac{\sum_{X_i \in N_k(y)} \varpi_i' \left\|X_i - y\right\|^r}{m_0 \sum_{i=1}^n \varpi_i}\right)^{1/r}, \tag{37}$$

where the notations $f_X$ and $\hat{d}_{X,m_0}$ are to clarify the dependency of $f$ on $X$. And from the condition, $\hat{d}_{m_0}(y) > \hat{d}_{m_0}(z)$ holds for all $z \in \varsigma$. Hence for sufficiently small $\epsilon > 0$ and for any $Z' = \{Z_1, \ldots, Z_n\}$ with $\left\|Z_j - X_j\right\| < \epsilon$, (37) becomes

$$f_Z(\varsigma) = \hat{d}_{Z,m_0}(y) = \left(\frac{\sum_{X_i \in N_k(y)} \varpi_i' \left\|Z_i - y\right\|^r}{m_0 \sum_{i=1}^n \varpi_i}\right)^{1/r}. \tag{38}$$

Hence by differentiating (38), the derivative of $f$ with respect to $X$ is calculated as

$$\begin{aligned}
\frac{\partial f(\varsigma)}{\partial X_j} &= \left(\frac{\sum_{X_i \in N_k(y)} \varpi_i' \left\|X_i - y\right\|^r}{m_0 \sum_{i=1}^n \varpi_i}\right)^{\frac{1}{r}-1} \times \frac{\varpi_j' \left\|X_j - y\right\|^{r-2}(X_j - y)I(X_j \in N_k(y))}{m_0 \sum_{i=1}^n \varpi_i} \\
&= \frac{\varpi_j' \left\|X_j - y\right\|^{r-2}(X_j - y)I(X_j \in N_k(y))}{\left(\hat{d}_{m_0}(y)\right)^{r-1} m_0 \sum_{i=1}^n \varpi_i}.
\end{aligned}$$

## E.9 Proof of Proposition D.2

From (5), note that for any $y \in \varsigma$, $\hat{d}_{m_0}(y)$ is expanded as

$$\hat{d}_{m_0}(y) = \left( \frac{\sum_{X_i \in N_k(y)} X_i' \left\| Y_i - y \right\|^r}{m_0 \sum_{i=1}^n X_i} \right)^{1/r}, \tag{39}$$

where $k$ is such that $\sum_{Y_i \in N_{k-1}(y)} X_i < m_0 \sum_{i=1}^n X_i \leq \sum_{Y_i \in N_k(y)} X_i$, and $X_i' = \sum_{X_j \in N_k(y)} X_j - m_0 \sum_{j=1}^n X_j$ for one of $Y_i$'s that is $k$-th nearest neighbor of $y$ and $X_i' = X_i$ otherwise. Hence, by letting $y = \arg\max_{z \in \varsigma} \hat{d}_{m_0}(z)$ and applying to (39), the filtration function $f_X$ at simplex $\varsigma$ becomes

$$f_X(\varsigma) = \hat{d}_{X,m_0}(y) = \left( \frac{\sum_{X_i \in N_k(y)} X_i' \left\| Y_i - y \right\|^r}{m_0 \sum_{i=1}^n X_i} \right)^{1/r}, \tag{40}$$

where the notations $f_X$ and $\hat{d}_{X,m_0}$ are to clarify the dependency of $f$ on $X$. And from the condition, $\hat{d}_{m_0}(y) > \hat{d}_{m_0}(z)$ holds for all $z \in \varsigma$. Hence for sufficiently small $\epsilon > 0$ and for any $Z' = \{Z_1, \ldots, Z_n\}$ with $\|Z_j - X_j\| < \epsilon$, (40) becomes

$$f_Z(\varsigma) = \hat{d}_{Z,m_0}(y) = \left( \frac{\sum_{X_i \in N_k(y)} Z_i' \left\| Y_i - y \right\|^r}{m_0 \sum_{i=1}^n Z_i} \right)^{1/r}. \tag{41}$$

Hence by differentiating (41), the derivative of $f$ with respect to $X$ is calculated as

$$\frac{\partial f(\varsigma)}{\partial X_j}$$

$$= \frac{1}{r} \left( \frac{\sum_{X_i \in N_k(y)} X_i' \left\| Y_i - y \right\|^r}{m_0 \sum_{i=1}^n X_i} \right)^{\frac{1}{r}-1} \times$$

$$\frac{\left\| Y_j - y \right\|^r I(Y_j \in N_k(y))\left(m_0 \sum_{i=1}^n X_i\right) - m_0\left(\sum_{X_i \in N_k(y)} X_i' \left\| Y_i - y \right\|^r\right)}{\left(m_0 \sum_{i=1}^n X_i\right)^2}$$

$$= \frac{\left\| Y_j - y \right\|^r I(Y_j \in N_k(y)) - m_0\left(\hat{d}_{m_0}(y)\right)^r}{r\left(\hat{d}_{m_0}(y)\right)^{r-1} m_0 \sum_{i=1}^n X_i}.$$

# F    Guideline for choosing TDA parameters

`PLLay` has several TDA parameters to choose: $K_{\max}$, $T_{\min}$, $T_{\max}$, $m$, and $m_0$ if DTM filtration is used. One can try grid search but it could be too time-consuming. More affordable approach is to compute the DTM filtration and the persistence diagram for some data and choose appropriate parameters that can reveal the topological and geometrical information of the data. Figure 5 illustrates one example of the digit $8$ in `MNIST` data. Figure 5(a) shows the contour plot of the chosen data.

When using a DTM filtration, we need to choose $m_0$ first. DTMs with different $m_0$ values extract different topological and geometrical information. When $m_0$ is small, a DTM filtration aggregates the data more locally, and the geometrical and homological information formed from the local structure is extracted. When $m_0$ is large, a DTM filtration aggregates the data more globally, and the geometrical and homological information formed from the global structure is extracted. From the digit $8$, we would first like to see the two-loop structure. And if we choose $m_0 = 0.05$, then as can be seen in Figure 5(b) and (c), the $1$st persistent homology extracts the two-loop structure, which is more directly expected from the contour plot of the data itself in Figure 5(a). However, if we choose $m_0 = 0.2$, then as can be seen in Figure 5(d) and (e), the two-loop structure disappears, since the two-loop structure is coming from more local geometry of the data. Meanwhile, as the DTM filtration aggregates the data more globally, the global geometry information that three points on the digit 8(top, center, bottom) being close to neighboring points and being centers of local clusters is extracted in the 0th persistent homology. For `MNIST` data, DTM filtrations with $m_0 = 0.05$ and $m_0 = 0.2$ extract different topological and geometrical information of the data. Hence for `MNIST` data, we used two parallel `PLLay`s with $m_0 = 0.05$ and $m_0 = 0.2$, respectively.

After choosing $m_0$, choosing other TDA parameters $K_{\max}$, $T_{\min}$, $T_{\max}$, $m$ is more straightforward. One can choose parameters so that the desired topological features are well extracted in the landscape. For $m_0 = 0.05$, as can be seen from Figure 5(c), choosing $K_{\max} = 2$, $T_{\min} = 0.06$, $T_{\max} = 0.3$, $m = 25$ will extract two 1-dimensional features of the persistence diagram in the corresponding landscape. For $m_0 = 0.2$, as can be seen from Figure 5(e), choosing $K_{\max} = 3$, $T_{\min} = 0.14$, $T_{\max} = 0.4$, $m = 27$ will extract two 1-dimensional features of the persistence diagram in the corresponding landscape.

# G    Experiment Details.

All the experiments were implemented using `GUDHI` The GUDHI Project [2020] and `Tensorflow` library in Python and `TDA` package Fasy et al. [2014] in R. We use mean and standard deviation across 20 runs of simulations with different network initializations. We remark that the basic purpose of our experiment design is to highlight the prospects and possibilities of using topological layer, not to win state-of-the-art performances.

(a) Digit 8 in MNIST data.

(b) Contour plot of DTM filtration, $m_0 = 0.05$.

(c) Persistence Diagram of DTM filtration, $m_0 = 0.05$.

(d) Contour plot of DTM filtration, $m_0 = 0.2$.

(e) Persistence Diagram of DTM filtration, $m_0 = 0.2$.

Figure 5: One example of the digit 8 in MNIST data, its contour plots and persistence diagrams of DTM filtration at $m_0 = 0.05$ and $m_0 = 0.2$. When $m_0 = 0.05$, DTM filtration aggregates more locally, and the 1st persistent homology extracts two loop structures of the digit 8. When $m_0 = 0.2$, DTM filtration aggregates the digit 8 more globally, and the 0th persistent homology extracts three connected component structures of the digit 8.

| | Corruption and noise probability | | | | | | | |
|---|---|---|---|---|---|---|---|---|
| | 0.00 | 0.05 | 0.10 | 0.15 | 0.20 | 0.25 | 0.30 | 0.35 |
| MLP | 0.8683 | 0.8425 | 0.8133 | 0.7850 | 0.7441 | 0.6997 | 0.6514 | 0.5732 |
| | (0.0063) | (0.0061) | (0.0087) | (0.0086) | (0.0098) | (0.0090) | (0.0124) | (0.0155) |
| MLP+S | 0.8597 | 0.8322 | 0.8060 | 0.7749 | 0.7364 | 0.6844 | 0.6372 | 0.5637 |
| | (0.0087) | (0.0086) | (0.0152) | (0.0147) | (0.0177) | (0.0187) | (0.0213) | (0.0161) |
| MLP+P | **0.8791** | 0.8538 | 0.8227 | 0.7910 | 0.7511 | 0.7045 | 0.6507 | 0.5753 |
| | **(0.0062)** | (0.0061) | (0.0103) | (0.0121) | (0.0109) | (0.0087) | (0.0120) | (0.0135) |
| CNN | 0.8506 | 0.8367 | 0.8030 | 0.7872 | 0.7541 | 0.7315 | 0.6778 | 0.6245 |
| | (0.0261) | (0.0246) | (0.0315) | (0.0340) | (0.0319) | (0.0447) | (0.0506) | (0.0478) |
| CNN+S | 0.8544 | 0.8058 | 0.7988 | 0.7938 | 0.7649 | 0.7055 | 0.6884 | 0.6281 |
| | (0.0194) | (0.1081) | (0.0252) | (0.0326) | (0.0215) | (0.1268) | (0.0372) | (0.0407) |
| CNN+P | 0.8790 | **0.8541** | **0.8364** | **0.8209** | **0.7855** | **0.7551** | **0.7044** | 0.6355 |
| | (0.0151) | **(0.0218)** | **(0.0214)** | **(0.0217)** | **(0.0247)** | **(0.0289)** | **(0.0230)** | (0.0404) |
| CNN+P(i) | 0.8635 | 0.8391 | 0.8113 | 0.7985 | 0.7671 | 0.7391 | 0.6841 | **0.6364** |
| | (0.0189) | (0.0153) | (0.0250) | (0.0275) | (0.0179) | (0.0302) | (0.0936) | **(0.0355)** |

Table 2: Test accuracy in `MNIST` experiments. In each cell, the top number corresponds to the average accuracy of the model at the corruption and noise probability, and the bottom number corresponds to the 1 standard deviation of the accuracies. At each column, the model with the best accuracy is bolded.

## G.1  MNIST handwritten digits.

For MNIST handwritten digits, we use `MNIST` dataset. Raw input data is a 784 dimensional vector (reshaped from 28 by 28) of real values, each value being the pixel intensity. We use 1000 random samples for the training set and 10000 samples for the test set. Cross-entropy loss was used to train the network for 100 epochs, using Adam optimizer with mini-batches of size 16.

**Topological layer.** For MLP+P and CNN+P(i), we use two parallel `PLLay`s at the beginning of MLP and CNN models with 32 nodes each and affine transformation, which are concatenated to the raw input to either MLP or CNN. We used the empirical DTM filtration in (5), where we define fixed $28 \times 28$ points on grid on $[-1, 1]^2$ and use $X$ as a weight vector for the fixed points. For one `PLLay`, we used $m_0 = 0.05$, $K_{\max} = 2$, $T_{\min} = 0.06$, $T_{\max} = 0.3$, $m = 25$, and for the other `PLLay`, we used $m_0 = 0.2$, $K_{\max} = 3$, $T_{\min} = 0.14$, $T_{\max} = 0.4$, $m = 27$. For CNN+P, we additionally use one `PLLay` after the convolutional layer, with $K_{\max} = 3$, $T_{\min} = 0.05$, $T_{\max} = 0.95$, $m = 18$.

**Baselines.** For the baselines, models were designed to have simple structures for quick comparisons:

- Vanilla MLP: one hidden layer with 64 units with ReLU activations.
- CNN: two convolution layers followed by two fully connected layers.
- SLay: for comparison with `PLLay`, two SLays are used with 10 nodes each, which are concatenated to the raw input to either MLP or CNN. We used the value $\nu = 0.005$ and $\nu = 0.01$ for the hyperparameter of each SLay, respectively.

**Result.** The Accuracy results for `MNIST` data in Figure 4 is represented with 1 standard errors in Table 2 and Figure 6. In Figure 6, the results for MLP, MLP+S, MLP+P are in Figure 6(a), and the results for CNN, CNN+S, CNN+P, CNN+P(i) are in Figure 6(b). We can see that `PLLay` consistently improves the accuracies of all baselines. In particular from Table 2 and Figure 6(b), the improvement on CNN is $1.7\% \sim 2.8\%$ when the corruption and noise is $0\% \sim 5\%$, and then the improvement goes up to $3.3\%$ when the corruption and noise becomes $10\% \sim 15\%$, and then starts to decrease as the corruption and noise further increases. As discussed in Section 5, this is because although the DTM filtration can robustly capture homological signals up to a moderate amount of corruption and noise, as seen in Figure 2, when the corruption and noise become too much, the topological structure starts to dissolve in the DTM filtration. Also, the accuracies for CNN+P are consistently higher than the accuracies for CNN+P(i), meaning that adding `PLLay` in the middle of the network indeed further improves the accuracy.

(a) Test accuracy in `MNIST` data for MLP, MLP+S, MLP+P.

(b) Test accuracy in `MNIST` data for CNN, CNN+S, CNN+P, CNN+P(i).

Figure 6: Test accuracy in `MNIST` experiments. `PLLay` contributes to consistent improvement in accuracy and robustness against noise and corruption. In particular, the improvement on CNN increases up to the moderate level of corruption and noise ($\sim 15\%$), and then start to decrease.

| | Noise probability | | | | | | | |
|---|---|---|---|---|---|---|---|---|
| | 0.00 | 0.05 | 0.10 | 0.15 | 0.20 | 0.25 | 0.30 | 0.35 |
| MLP | 0.2000 (0.0014) | 0.2001 (0.0031) | 0.1997 (0.0020) | 0.1994 (0.0029) | 0.1998 (0.0009) | 0.2003 (0.0010) | 0.2004 (0.0016) | 0.1999 (0.0011) |
| MLP+S | 0.2054 (0.0126) | 0.2028 (0.0129) | 0.2171 (0.0364) | 0.2171 (0.0364) | 0.2121 (0.0236) | 0.2159 (0.0301) | 0.2115 (0.0193) | 0.2057 (0.0180) |
| MLP+P | 0.8082 (0.0103) | 0.7906 (0.0082) | 0.7660 (0.0115) | 0.7456 (0.0104) | 0.7181 (0.0100) | 0.6942 (0.0130) | 0.6545 (0.0110) | 0.6218 (0.0102) |
| CNN | 0.9466 (0.0116) | 0.9247 (0.0152) | 0.9053 (0.0195) | 0.8791 (0.0255) | 0.8224 (0.1474) | 0.8323 (0.0298) | 0.7963 (0.0331) | 0.7401 (0.1293) |
| CNN+S | 0.9412 (0.0182) | 0.8881 (0.1612) | 0.8142 (0.1900) | 0.8142 (0.1900) | 0.8197 (0.1473) | 0.7777 (0.1875) | 0.6580 (0.2622) | 0.7195 (0.1778) |
| CNN+P | **0.9511** **(0.0140)** | 0.9249 (0.0308) | **0.9095** **(0.0329)** | **0.8941** **(0.0305)** | **0.8619** **(0.0366)** | **0.8480** **(0.0173)** | **0.8087** **(0.0396)** | **0.7668** **(0.0319)** |
| CNN+P(i) | 0.9449 (0.0343) | **0.9319** **(0.0290)** | 0.8965 (0.0471) | 0.8873 (0.0143) | 0.8577 (0.0349) | 0.8285 (0.0515) | 0.7954 (0.0516) | 0.7543 (0.0553) |

Table 3: Test accuracy in ORBIT5K experiments. In each cell, the top number corresponds to the average accuracy of the model at the noise probability, and the bottom number corresponds to the 1 standard deviation of the accuracies. At each column, the model with the best accuracy is bolded.

## G.2 Orbit recognition.

For orbit recognition, we use ORBIT5K dataset [Adams et al., 2017, Carrière et al., 2020], a synthetic dataset used as a benchmark in Topological Data Analysis. It consists of a point cloud generated by the following discrete dynamical system: given an initial point $(x_1, y_1) \in [0,1]^2$ and a parameter $r > 0$, we generate a point cloud $\{(x_n, y_n) \in [0,1]^2 : n = 1, \ldots, N\}$ as

$$\begin{cases} x_{n+1} = x_n + r y_n (1 - y_n) & \mod 1, \\ y_{n+1} = y_n + r x_{n+1} (1 - x_{n+1}) & \mod 1. \end{cases}$$

For comparison with Adams et al. [2017], Carrière et al. [2020], we use parameters $r = 2.5, 3.5, 4.0, 4.1, 4.3$, with random initialization of $(x_1, y_1)$ and $N = 1000$ points in each simulated orbit. We generated 1000 orbits per each value of $r$, and randomly split the 5000 observations in $70\% - 30\%$ training-test sets as in Carrière et al. [2020]. Cross-entropy loss was used to train the network for 100 epochs, using Adam optimizer with mini-batches of size 16. For the noiseless case, the experiment for PointNet is repeated 5 times, and the experiment result for PersLay is from Carrière et al. [2020].

**Topological layer.** For MLP+P and CNN+P(i), we use one PLLay at the beginning of MLP and CNN models with 64 nodes and affine transformation, which is solely used as the input to MLP or concatenated to the raw input to CNN. We used the empirical DTM filtration in (4), where we define fixed $40 \times 40$ points on grid on $[0.0125, 0.9875]^2$ and use $X$ as the empirical data points. We used $m_0 = 0.01$, $K_{\max} = 2$, $T_{\min} = 0.03$, $T_{\max} = 0.1$, $m = 17$. For CNN+P, we additionally use one PLLay after the convolutional layer, with $K_{\max} = 2$, $T_{\min} = 0.05$, $T_{\max} = 0.95$, $m = 18$.

**Baselines.** For the baselines, models were designed to have simple structures for quick comparisons:

- Vanilla MLP: one hidden layer with 32 units with ReLU activations.
- CNN: two convolution layers followed by two fully connected layers.
- SLay: for comparison with PLLay, one SLay is used with 16 nodes, which is concatenated to the raw input to either MLP or CNN. We used the value $\nu = 0.01$ for the hyperparameter of SLay.

**Result.** The accuracy results for ORBIT5K data in Figure 4 is represented with 1 standard errors in Table 3 and Figure 7. In Figure 7, the results for MLP, MLP+S, MLP+P are in Figure 7(a), and the results for CNN, CNN+S, CNN+P, CNN+P(i) are in Figure 7(b). From Figure 7(a), we observe that PLLay improves over MLP and MLP+S by a huge margin ($42\% \sim 60\%$). In particular, without PLLay, MLP and MLP+S remain at random classifiers, which implies that the topological information is indeed critical for ORBIT5K. In Figure 7(b), PLLay improves over CNN or CNN+S consistently

as well. Moreover, due to the high complexity of `ORBIT5K`, CNN suffers from high variance at corruption and noise probability $0.2, 0.35$, while `PLLay` can effectively reduce the variance at those simulations and make the models more stable by utilizing robust topological information from the DTM function. Also, the accuracies for CNN+P are almost always higher than the accuracies for CNN+P(i), meaning that adding `PLLay` in the middle of the network indeed further improves the accuracy.

(a) Test accuracy in `ORBIT5K` data for MLP, MLP+S, MLP+P.

(b) Test accuracy in `ORBIT5K` data for CNN, CNN+S, CNN+P, CNN+P(i).

Figure 7: Test accuracy in `ORBIT5K` experiments. `PLLay` contributes to consistent improvement in accuracy and robustness against noise and corruption. In particular in (b), when the corruption and noise probability is $0.1, 0.25, 0.35$, `PLLay` effectively reduces the variance of classification accuracy.