[Reviews · NeurIPS 2020]

Review 1

Summary and Contributions: This paper describes a novel layer for neural networks that, being based on persistence landscapes (a functional summary of topological features), are capable of making use of the inherent topology of data sets in order to improve classification performance, for example. The paper shows that the new layer is differentiable and can be employed at any position of a neural network, making it highly versatile. Experiments on numerous data sets demonstrate the improvements provided by the new layer.

Strengths: The main strength of the paper lies in the conceptual simplicity of the approach: persistence landscapes are a well-established tool in topological data analysis (TDA) and their use in standard deep learning architectures is well appreciated. The paper explains the use of these concepts in a clear and approachable manner (apart from some minor issues, which I shall subsequently discuss). I envision that disseminating more information about TDA methods will be beneficial for the machine learning community at large, even more so as the proposed neural network layer potentially gives rise to new hybrid applications (i.e. applications that *integrate* topological features into existing architectures).

Weaknesses: While I feel favourably about this paper, there are still some issues that prevent me from fully endorsing it. My primary reservation is that the experimental or practical benefits are clearly demonstrated and that the experimental setup and 'presentation' is somewhat shallow: - When depicting accuracies in Figure 3, please show the standard deviations alongside the accuracies (as additional error bars, for example) in order to make the methods comparable. It seems to me that the achieved gains for MNIST are not necessarily significantly better than other methods, but I could be wrong here (I do appreciate the setup in terms of noise or 'corruption' levels, though; I think more of these kinds of experiments are always helpful in ML!) - In addition to a better way of reporting the results, I would also like to see a more detailed comparison to baseline methods. I understand that the natural comparison partner consists of *other* topological layers. However, next to these neural network baseline, I would also suggest taking a look at kernels for persistence diagrams or metrics between them (though I could understand that the latter type of functions might not be efficiently computable). However, there are many kernel functions that could be equally well used here: - Carrière et al.: Sliced Wasserstein Kernel for Persistence Diagrams, ICML 2017 - Kusano et al.: Persistence weighted Gaussian kernel for topological data analysis, ICML 2016 - Reininghaus et al.: A Stable Multi-Scale Kernel for Topological Machine Learning, CVPR 2015 In a similar vein, the persistence landscape formulation also affords a kernel formulation; it would be interesting to see how the proposed method compares to this. The persistence image method would also be a natural comparison partner, in case the kernels turn out to be computationally problematic. I think this is necessary in order to see how much accuracy is due to the 'switch' from more classical models (e.g. SVMs) to 'deep architectures'. - The experimental section would also be stronger if the comparison in terms of architectures would be exactly the same. Since a CNN architecture already has a higher baseline accuracy than `PersLay`, for example, would it not be better to try to disentangle the change in architecture from the change in layers?

Correctness: The paper and all of its derivations are technically correct, as far as I can tell. I have a few minor issues concerning terminology and phrasings, though: - I would suggest using 'persistent homology calculation' or 'persistent homology functor' instead of just 'persistent homology' (for example in the abstract). - The filtration in l. 74 could also be defined using `\subseteq` to my understanding. It is not necessary that the simplicial complexes are proper subsets of each other. - Persistence diagrams are typically defined over $\mathbb{R} \times \mathbb{R} \cup \{\infty\}$ because the creation values are always finite.

Clarity: The paper is very accessible for the most part. There are some formulations that could be improved, though: - the sentence in l. 51-- '...differentiability is not guaranteed...' could be rephrased; does it refer to the theoretical requirements for filtrations to be differentiable (i.e. injectivity, tameness) - 'in the way that we suffer less from extreme...' --> 'in a way that makes it less prone to extreme...' - What does the sentence in l. 87 mean? Is it missing a 'respectively' at the end? - Consider typesetting the equation $P_n(x)$ in l. 113 as a stand-alone equation. Currently, it is hard to read. Moreover, what are typical choices for the DTM filtration and its parameters? How is $m_0$ chosen typically? I think it would be useful to provide an example here. - In Eq. 3 and Eq. 4, the comma belongs in the `equation` environment - I would remove the mention of 'homology group' in l. 148, since it was not defined in the paper (and moreover, it is not required to understand the respective sentence) - 'intialized as equal weight' --> 'initialized weights uniformly'

Relation to Prior Work: Related work is adequately referenced for the most part. The authors may want to consider citing the kernels mentioned above. Moreover, in the context of differentiable loss terms, the following work might be relevant: M. Moor et al. Topological Autoencoders https://arxiv.org/abs/1906.00722

Reproducibility: Yes

Additional Feedback: Here are some additional suggestions to help improve the paper: - 'topological theories' --> 'topological concepts' - 'proposed topology loss term' --> 'proposed a topological loss term' - 'they lack stability result' --> 'they lack stability results' - 'we define empirical DTM' --> 'we define the empirical DTM' - For some of the maps, consider using `\colon` instead of `:` in order to fix the spacing (`:` is used as an operator, whereas `\colon` is used as a definition) - In the appendix, there are some additional typos in equations; $m0$ is often used instead of $m_0$, for example. I would suggest checking this section again. - I would suggest sorting citations when citing multiple papers; for citations with multiple papers, alphabetically ordered citations would be easier to read. Alternatively, the authors may want to consider using a 'numbered' citation style (this would also free up additional space in the paper, which could be used to extend the experimental section, for example). - In some places, the paper uses 'raw' citations without brackets; this looks somewhat strange because the citations are part of a sentence. For example, in the introduction, the sentence in l. 25 'geometric way Carlsson [2009], Ghrist [2008]' is somewhat strange to read. Consider using `\citep` if `natbib` is used. --- Update after author response: I thank the authors very much for the rebuttal, which addressed my concerns. I am raising my score accordingly!


Review 2

Summary and Contributions: The paper provides a new topological layer, called LandLayer, that can be inserted anywhere inside the network with a differentiable version to embed into current DL frameworks. Contrary to previous works, the authors use Distance to Measure as the filtration function and construct persistence diagrams based on them. After obtaining the persistence diagrams, the authors further construct Persistence landscapes over them in order to embed into a fixed dimension vector space, since persistence diagrams are a multiset, thus making it difficult to use them directly with the standard feedforward and other DL layers. The authors then provide a standard mapping algorithm to map the outputs of the persistence landscapes to R, via a differentiable parametrized map g, as denoted in algorithm 1 on page 5 of the paper. As mentioned in this citation - [Chazal et al., 2011, 2016a] in the paper, the DTM provides a robust version of the distance function. The authors have also provided stability theorems and a tighter upper bound of their approach as compared to Hofer et al. [2017] (the first paper using topological signatures - Deep learning with topological signatures from NIPS 2017). DTM provides additional benefits of being used with arbitrary input data and is agnostic to the choice of the type of filtration performed, which makes it quite useful for diverse practical applications. Interestingly, as shown in figure 3, the standard deviations in the results are lower for MLP baselines.

Strengths: The method has decent novelty. Incorporating the DTM functions seems to work quite well based in the two experiments provided. Use of persistence landscapes for creating a fixed size mapping rather than persistence diagrams seems good. The empirical evaluation shows substantial improvements when the noise level is increased in the input data. Tighter bound and stability theorems compared to previous works introducing such topological layers, however a bit trivial. Can be fused with any existing standard DL pipeline and any type of layers.

Weaknesses: What is the computational complexity of computing a persistence landscape given a persistence diagram? This is crucial in understanding how much overhead this layer adds to the neural network's training time. Section 3.3 seems quite straightforward. As already pointed out TF or Pytorch allows for automatic differentiation in their frameworks. What is h_{top}? Comparing against a vanilla 2-layer MLP or 2-layer CNN isn't really fair. Why aren't standard models like Resnet considered and shown to improve with the addition of your topolayer? In Line 263, when noise becomes too much then the topological structure is lost, so then how does it compare against a VAE denoising layer that helps denoise the data before feeding to a CNN? E.g. https://www.pyimagesearch.com/2020/02/24/denoising-autoencoders-with-keras-tensorflow-and-deep-learning/ There are several state-of-the-art denoising networks one can use in conjunction with a CNN to achieve robustness to noise, which are not compared against in this work. E.g: https://www.groundai.com/project/dn-resnet-efficient-deep-residual-network-for-image-denoising/1 The claim of “improving generalization” (Line 235) is unsubstantiated because there are no theoretical or empirical results to justify such an improvement. The experimental evaluation is extremely limited, on just using two datasets. Also, the authors could have used more baselines. Using more challenging datasets such as CIFAR-10 from image domain, performing graph classification, more challenging datasets from the Point Clouds could substantially justify the results. The method seems to be a bit slow in practice (based on what I observed from their code). It would be great to provide runtimes for various baselines. Essentially, the authors should provide a complexity analysis and provide empirical runtime values. The authors have stated that they intentionally used smaller training set size. However, there should be experiments performed on the whole dataset as well, maybe in the supplementary section, in order to verify how their method performs in the limiting scenario when the whole training dataset is used. This makes it all the more necessary to include several datasets including more complicated ones. The authors could have also evaluated networks with explicit regularization techniques in order to verify how their proposed topological layer performs in comparison to contemporary neural networks that are enhanced with such a layer. A big question remains on the performance of their proposed layer. Extensive experiments as done in this paper - http://proceedings.mlr.press/v97/hofer19a/hofer19a.pdf, would have been better. The paper was a bit difficult to follow at times - how to apply the DTM on the input, especially the fixed grid point method in equation 4. I feel the authors could have provided detailed explanations for this in the supplementary section.

Correctness: yes

Clarity: yes

Relation to Prior Work: yes

Reproducibility: No

Additional Feedback:


Review 3

Summary and Contributions: I thank the authors for the responses. I still feel that PersLay could be used for MNIST combined with DTM. Nevertheless, overall, I am positive about this paper's contribution (as stated below) and keep my score. Persistent homology based feature representations have recently been used successfully in summarizing complex data. While persistent homology produces useful multiscale summaries for complex data, the resulting representation, the so-called persistence diagram (which is a multiset of points), are not easy to use with deep learning. Recent years have seen a new trend in developing a differentiable ``topological layer" to process persistence summaries. This paper proposes a topological layer, called LandLayer, based on the weighted persistence landscape representation of persistence diagrams. (The precise representation will be learned during backpropagation.) This layer is differentiable to input, and thus can be inserted anywhere in a deep learning network. A specific instantiation of this layer is by using the DTM function as filtration to generate persistence diagrams. The paper also provides stability studies (including for the case when DTM is used). Some empirical studies are provided to show the effectiveness of the proposed LandLayer.

Strengths: + Having an easy to use topological layer is important, and will help significantly broaden the use of topological summaries with deep learning. The paper is tackling an important problem. + The concept of topological layer and key components so as to allow differentiating such a layer have been around. But I think this paper is the first to explicitly formulate to differentiate it w.r.t. arbitrary input, which further allows to put this layer anywhere in a deep neural network (e.g, the input can be output from previous layer). I consider this to be a key contribution. + The stability analysis is nice. + The instantiation with DTM function is useful -- I think one can use other choices of filtrations. But DTM probably works in many cases and can serve as a good default setup for a blackbox application of the proposed LandLayer. While I will mention some weakness below, I think the generality of this LandLayer, its presumably easy applicability to any NN, as well as the theoretical analysis to me are very solid contribution, and thus I am overall positive of the submission.

Weaknesses: - The paper is a little hard to read. The contributions are somewhat mixed in the exposition, so it is hard to decouple and identify exactly what's new. For example, the paper mentioned that differentiability is guaranteed for arbitrary input in their setting, while not the case in prior work. But it is not clear to me whether there is any fundamental reason why previous approach cannot do it, or they just didn't formulate it that way. What is the key idea in their formulation to achieve now? Also, while this paper uses persistence landscapes, any reason other representations will not work? How exactly does this work compare with PersLay ("PersLay: a neural network layer for persistence diagrams and new graph topological signatures", AISTATS 2020)? - The empirical studies could be improved. (a) For the first set of experiments on MNIST, any reason why one does not combine CNN with PersLay (instead of just using SLayer) ? (b) For the second ORBIT5K data, is the same filtration used for all the comparison methods? (Otherwise, the improvement could be from DTM filtration.) (c) More thorough experimentation over more data sets would be good.

Correctness: From what I see in the main text, all looks good. I didn't read the Supplement carefully.

Clarity: Overall it is okay, although sometimes the key new ideas are not clarified and emphasized.

Relation to Prior Work: See comments above in "weakness".

Reproducibility: Yes

Additional Feedback:

[Author Response · NeurIPS 2020]

**Reviewer #1**

**a.** We actually have presented the average standard deviation (sd) across different noise/corruption rates for each method in the bottom of Figure 3, as well as in pages 23, 24 of the appendix where accuracies with error bars are shown in full (for the sake of saving space, only the average sd's are shown in the main text). We will make this more clear.

**b.** As you pointed out, the improvement is marginal in MNIST ($2 \sim 3\%$ on average). MNIST is used for the pedagogical purpose as its topological feature can be seen clearly both by the naked eye and the persistence diagram. In more complex ORBIT5K, we achieved significant improvement both in accuracy and variance reduction over baselines, whereas the state-of-the-art classifier PointNet which is not using the topological information is much worse. We conjecture the degree of improvement is related to how much the topological information is important in the data.

**c.** We appreciate your suggestion on using kernels for persistent homology for comparative purposes. Indeed, the logarithmic transformation for our choice of $g_\theta$ in our paper (174) is a special case of Gaussian kernel for the persistence diagram, which didn't bring much difference compared to the affine transformation (172) in our simulations. Moreover, 1) while our method gives an explicit differentiability guarantee (sec 3.3), it wouldn't be straightforward (or at least require substantial extra work) for other kernel featurization methods so they are not directly compatible in our setting, and 2) in ORBIT5k experiment our method appears to beat the best result of PersLay which utilizes flexible kernel functions. That being said, connection to applying kernels for persistence diagrams sounds very interesting; indeed weighted Gaussian kernel has very similar properties to Landscapes. We will elaborate on this in the discussion section.

**Reviewer #2**

**a,g.** Computational complexity depends on how LandLayer is used. Computing the DTM is $O(n + m \log n)$ where $n$ is the input size and $m$ is the grid size. Computing persistence diagram is $O(m^{2+\epsilon})$ for any small $\epsilon > 0$ when we choose the simplicial complex $K$ in line 129 to grow linear with respect to the grid size such as cubical complex or alpha complex. Computing the persistence landscape grows linearly with respect to the number of homological features in the persistence diagram, which is the topological complexity of the input and do not necessarily depend on $n$ or $m$. For our experiments, we consider fixed grids of size $28 \times 28$ and $40 \times 40$ as in line 649 and 686, so the computation is not heavy. Also, if we put LandLayer only at the beginning of the deep learning model, then LandLayer can be pre-computed and needs not be calculated at every epoch in the training. We will add this discussion in Section 6.

**b.** As in line 123, $h_{top}$ is the proposed topological layer equivalent to eq.(5). We will make this more clear. Furthermore, Sec 3.3 is to guarantee the differentiability of $h_{top}$, one of the main contributions of our paper. The automatic differentiations from tensorflow or pytorch are correct only if the function is *guaranteed to be differentiable*. Hence, checking the differentiability of $h_{top}$ is critical. Also, $h_{top}$ consists of the piecewise linear function $\lambda_k$, which is not differentiable at changepoints. The number of changepoints can be large, so automatic differentiation is inefficient and plugging in the explicit formula for the derivatives of $\lambda_k$ is better, which is done in the source code.

**c,f,h.** As stated in 22-24, our paper is more focused on proposing a comprehensive methodological framework with theoretical validation. Experiments are to show the enhanced learnability of neural networks with LandLayer under simple setup, not to build the best model for a particular dataset. Although LandLayer can lead to future applied works, those are beyond the scope of the paper. To get the state-of-the-art, one should design the architecture adaptively to the task, after a considerable amount of research on which model to use and where to place LandLayer. In Table 1, the simple CNN+LandLayer still beats the current state-of-the-art, but that's not our main goal of the paper. In a similar vein, since our method is designed to efficiently capture significant topological features, we intentionally used small sample size to verify the efficiency of our method (242-243), which is another benefit of using topological layer; we conjecture for very large sample size, the gap might be shrinking to some extend. Also please see **b** to Reviewer #1.

**d,i.** As in **c**, the experiment is to verify the enhancement by adding LandLayer but not to beat existing approaches. LandLayer utilizes topological features, which not only are robust to noise but also provide information that is not extractable by existing methods. As in Table 1, even without the noise, the state-of-the-art PointNet is performing badly due to lack of the topological information. Also, as in line 286-287, LandLayer can be placed anywhere in the deep learning network, and hence it can be combined with denoising layer or regularization techniques as well.

**e.** We just meant to say improving the overall accuracy by adding the proposed LandLayer, as verified in the result.

**j.** Equation (4) is the direct application of (2) where $X_i$'s are fed into the place of $\varpi_i$'s and $Y_i$'s into the place of $X_i$'s.

**k.** Regarding reproducibility, we have provided all the details in Section D.9 of the appendix and the full source code too. We do not see any issues with the reproducibility of our experimental result. Can you please elaborate on this?

**Reviewer #3**

**a.** As you pointed out, some recent work proposed their topological layers as in Hofer et al 2017, Carrière et al 2020 (PersLay), etc. But their differentiability is only guaranteed w.r.t the persistence diagram, NOT the layer input. So they can't be placed in the middle of the network, but only in the beginning of the network. To the best of our knowledge, this general differentiability guarantee, not to mention other favorable properties, has not yet appeared in the literature.

**b.** PersLay used their own persistent homology (i.e., extended diagram) and their own filtration functions, both of which characterize PersLay itself. So we didn't use DTM for PersLay but still reported it for ORBIT5K for comparative purpose as it gives the state-of-the-art result. Moreover, for SLayer we used the exactly same DTM filtration. Also see **c** to Reviewer #2.

[Meta-Review · NeurIPS 2020]

The reviewers are mostly satisfied by the author response, and agree about acceptance. There is still a remaining concern that additional baselines/experiments would be desirable, but given the theoretical contributions and the layer's differentiability guarantees, the contribution is already considered sufficient to warrant acceptance.